



# Seasonal evolution of the subglacial hydrologic system beneath the western margin of the Greenland Ice Sheet inferred from transient speed-up events

Grace Gjerde[1], Mark D. Behn[1], Laura A. Stevens[2], Sarah B. Das[3], and Ian Joughin[4]

[1]Department of Earth and Environmental Sciences, Boston College, Chestnut Hill 02467, United States
[2]Department of Earth Sciences, University of Oxford, Oxford, OX1 2JD, United Kingdom
[3]Department of Geology and Geophysics, Woods Hole Oceanographic Institution, Woods Hole, MA 02543, United States
[4]Polar Science Center, Applied Physics Laboratory, Seattle, WA 98105, United States

*Correspondence to*: Grace Gjerde (gjegp@sbcglobal.net)

**Abstract.** The transport of meltwater from the surface to the bed of the Greenland Ice Sheet is well understood to result in elevated surface velocities, although this relationship remains poorly resolved on a seasonal scale. Transient speed-ups associated with supraglacial lake drainages, which generally occur in the early- to mid-summer melt season, have been studied in detail. However, the connection between basal hydrology and ice dynamics is less well understood in the late melt season, after most lakes have
ceased draining and meltwater input to the bed is through widely distributed moulins. Here, we use a Global Positioning System (GPS) array to investigate transient speed-up events in response to runoff across the 2011 and 2012 melt seasons and use these data to infer the evolution of subglacial conditions beneath the ice sheet in western Greenland. We find no relationship between the magnitude of runoff and the amplitude of speed-up events; we do observe a general trend of increasing velocity responses and decreasing variability in the velocity response across the GPS array as the melt season progresses. Early-season velocity transients
(frequently associated with lake drainages) produce highly variable speed-up and pronounced uplift across the array. Late-season melt events produce longer, higher amplitude, and more uniform velocity responses, but do not produce large or coherent uplift patterns. We interpret our results to imply that by the late melt season, most subglacial channels and/or connective flow pathways between cavities are closing or have closed, sharply lowering basal transmissivity. At the same time, moulins formed throughout the melt season remain open, producing pervasive and widely distributed surface-to-bed pathways. The result is that small
magnitude melt events can rapidly supply meltwater to the bed and overwhelm the subglacial system, decreasing frictional coupling. This response contrasts with early-season lake drainage events when surface-to-bed pathways are not yet open, and therefore, similarly small magnitude melt events do not have the same impact. Finally, we show that due to their extended duration and amplitude, late-season melt events accommodate a larger fraction of the annual ice motion than early-season lake drainages but their net influence on ice sheet motion remains small (2–3% of annual displacement).


## 1 Introduction

The rate of mass loss from the Greenland Ice Sheet is accelerating (Hanna et al., 2024) due to a combination of increased surface
melting (van den Broeke et al., 2009; Hanna et al., 2013) and changes in ice-sheet flow dynamics (Pritchard et al., 2009; Schoof, 2010; Hewitt, 2013; Flowers et al., 2015). Global positioning system (GPS) observations of the western Greenland Ice Sheet show that ice velocities and surface melt increase in tandem on both seasonal (Zwally et al., 2002) and daily-to-weekly timescales (van del Wal et al., 2008; van de Wal et al., 2015). However, the details of the relationship between ice-sheet velocity and the input of



surface-melt to the bed is non-linear and varies throughout the melt season (Zwally et al., 2002; Bartholomew et al., 2010; Hoffman
et al., 2011; Stevens et al., 2016), reflecting changes in the subglacial hydrologic system and its coupling with basal sliding
conditions (Schoof, 2010; Hewitt, 2013). Untangling these feedbacks has led to challenges in predicting whether, and by how
much, ice-sheet velocities will change in a warming climate.

One avenue for improving our understanding of the subglacial hydrologic system is to study how the ice sheet responds to sudden
meltwater-input events that produce transient increases in ice-sheet sliding. These transient speed-ups can provide insights into
basal conditions that cannot be inferred from the background velocity curve alone. The most well-studied of these transient sliding
events are associated with rapid supraglacial lake drainages, which occur in the early-to-middle portion of the melt season. Along
the western margin of the Greenland Ice Sheet, summer melting results in the formation of supraglacial lakes, filling topographic
closed basins on the ice-sheet surface (Pitcher and Smith, 2019). These lakes, which range from 10s to 1000s of meters in diameter,
are observed to drain rapidly (<1–2 hours) via hydro-fractures that form beneath the lake basin (Das et al., 2008; Doyle et al., 2013;
Stevens et al., 2015; Chudley et al., 2019). Transient increases in ice-sheet surface velocity coincide with these drainages, as the
input of meltwater to the glacial bed reduces frictional contact between the ice and bedrock (Das et al., 2008; Andrews et al. 2014;
Andrews et al. 2018). During the lake drainage events, these velocity transients coincide with surface uplift caused by hydraulic
overpressure of the basal meltwater system (Das et al., 2008; Andrews et al. 2018). Early-season lake drainages tend to generate
sustained sliding anomalies, with the ice sheet remaining uplifted on timescales of days-to-weeks; mid-season lake drainage events
have shorter sliding anomalies with uplift confined to timescales of hours-to-days (Lai et al., 2021). These observations suggest
that the hydraulic transmissivity (i.e., the ability of the meltwater to move through the basal hydrologic system) becomes more
efficient beneath the lake as the melt season progresses (Lai et al., 2021), consistent with model predictions (Schoof, 2010) and
observations (Chandler et al., 2013; Andrews et al. 2014; Andrews et al. 2018) premised on a seasonal evolution towards a more
channelized subglacial meltwater system with increasing meltwater input (e.g., Schoof, 2010).

However, conflicting observations, such as increased basal uplift during periods of decreasing ice velocity (Andrews et al. 2018),
decreasing velocities while average moulin hydraulic head remains constant (Andrews et al. 2014), and, on decadal timescales,
declining annual velocities while annual melt increases (Tedstone et al., 2015), have led to multiple interpretations of the
mechanisms responsible for observed increases in subglacial efficiency. For example, changes in subglacial cavity connectivity
and their subsequent dewatering (Andrews et al., 2014; Tedstone et al., 2015; Hoffman et al., 2016; Andrews et al., 2018) and/or
sediment consolidation (Andrews et al. 2014; Clarke, 2005) have both been proposed as potential mechanisms for decreasing
velocities. Moreover, the exact timing of the transition from one state to another (e.g., cavities to channels or dilation to
consolidation of sediments) is difficult to determine. Ice thickness likely also plays a role, with basal channels creeping closed
more quickly under thick ice (Bartholomaus et al., 2011; Chandler et al., 2013; Doyle et al., 2014; Andrews et al., 2018), and
greater overburden pressures promoting sediment compaction (Clarke, 2005). These observations highlight the need for further
study on the evolution of basal conditions.

By the late melt season, infrequent supraglacial lake drainages prevent using lake drainage events to quantify subglacial hydrologic
conditions. However, transient ice-sheet speed-up events associated with melt or precipitation are observed late in the melt season
and, in some cases, occur after ice-sheet surface velocities have decreased below the average winter velocity (Joughin et al., 2013;
Andrews et al. 2018; Ing et al., 2024). Velocity increases driven by regional melt and precipitation events, contrast with those
driven by lake drainage or "flood" events because these velocity increases are produced by order-of-magnitude smaller surface-to-





bed meltwater fluxes. Studies investigating the transient response of the ice sheet to meltwater inputs in the late melt season report

contrasting findings on the importance of these melt events for annual ice-sheet velocities. Doyle et al. (2015) argued that late-season melt events could have a widespread influence on ice-sheet velocities in western Greenland. By contrast, more recent observations by Ing et al. (2024) suggest late-season melt events have a limited impact on annual ice velocities due to the relatively short durations of the melt events. Thus, the relationship between late-season melt events and basal transmissivity during this period is not well understood, and the lack of direct observations of these melt events limits our ability to make inferences about

system behavior. Moreover, it remains uncertain whether late-season melt or precipitation events contribute to ice-sheet acceleration at a magnitude comparable to that of lake drainages.

Here we analyze a series of late-season speed-up events recorded by a GPS array (Fig. 1) deployed near North Lake (68.66 °N, -49.52 °W; Das et al., 2008) located in the mid-ablation zone of the western margin of the Greenland Ice Sheet, roughly 50 km

south of the Sermeq Kujalleq (Jakobshavn-Isbrae) catchment. Using GPS observations of ice-sheet surface position in two melt seasons from 2011 and 2012, we use a Network Inversion Filter (NIF) (Stevens et al., 2015) to compare the velocity response of a late-season, transient speed-up event with the velocity response of an early-season lake drainages at the same location. Compared to the early-season lake drainage, we find that late-season speed-up event has larger speed-up amplitudes, more spatially uniform patterns of speed-up across the GPS array, and smaller uplift responses. To extend this analysis to examine the change in subglacial

conditions throughout the entire melt season, we characterize the absolute and relative speed-up magnitude and variability across the GPS array for transient speed-ups associated with both "local" lake drainages and "regional" melt events and compare these ice-response indicators with an estimate of the speed-up event runoff. Overall, we find no relationship between event speed-up and runoff magnitude; however, over the course of the melt season, we find an increase in the magnitude of speed-up relative to background ice velocities and a gradual reduction in the spatial variability of the ice-sheet response during an individual speed-up

event. These trends persist when considering only regional melt events occurring over the mid- to late- melt season. Finally, we interpret our results in the context of physical models for the evolution of the subglacial hydrologic system.

## 2 Methods


This study utilizes a GPS array and Regional Atmospheric Climate Mode (RACMO) runoff estimates in the ablation zone of the western Greenland Ice Sheet to capture transient speed-ups (Fig. 1). Below we describe the GPS data collection and processing, our application of the Network Inversion Filter (NIF), and our approach for defining and characterizing individual transient speed-up events. We also describe how the runoff for each speed-up event was estimated.


### 2.1 GPS-observed ice-sheet horizontal velocities and uplift

The North Lake basin is located at ~950 m above sea level (a.s.l.) (Fig. 1a) and is the field locality for several previous studies on supraglacial lake drainage that use ice-sheet surface position observations from a 16-receiver GPS array deployed from 2011 to

2014 (Stevens et al., 2015; 2016; 2024; Lai et al., 2021). Due to limited data availability in the latter two years of the deployment, we focus on the melt-season observations from primarily 14 receivers in 2011 and 12 receivers in 2012. The GPS data were collected continuously at 30-second resolution on dual frequency Trimble NetR9 receivers. The on-ice stations were individually processed as kinematic sites relative to the Greenland GPS Network (GNET) KAGA bedrock base station using the Track module



(Chen, 1998) within the GAMIT/GLOBK software package (Herring et al, 2010). The resulting 30-s horizontal position estimates

were used to calculate along-flow ice-sheet surface velocities using a sliding least-squares regression with a window width of 6 hours, following Stevens et al. (2016). The number of stations recording high-quality data varied through the deployment, with a maximum of 15 stations and a minimum of 11 stations available for characterizing a given speed-up event.

We used a Network Inversion Filter (NIF) algorithm (Segall and Mathews, 1997) developed for glaciological applications (Stevens

et al., 2015) to characterize the pattern of speed-up associated with a late-season melt event in 2011 and compared this pattern to that observed during a supraglacial lake drainage event at the same location earlier in the year. The NIF inverts timeseries of GPS ice-sheet surface positions for vertical hydro-fracture opening, sub-horizontal slip, and basal-cavity opening (Stevens et al., 2015), assuming the ice behaves as a homogeneous, elastic material (Okada, 1985; Segall, 2010). This assumption is valid for ice deformation during or immediately following a lake drainage or similar transient speed-up (Stevens et al., 2015; Lai et al., 2021).

The NIF quantifies the increased rate of ice flow relative to a background rate estimated from pre-speed-up event station velocities. For example, Stevens et al. (2015) utilized the NIF to investigate a series of early-season supraglacial lake drainages at North Lake from 2011–2013. For the 2011 lake drainage event on day of year (DOY) 169 (i.e., 2011/169), they reported a maximum surface uplift of 0.6 m and Lai et al. (2021) reported that ice-flow velocities remained elevated above their pre-drainage background rate for ~2 weeks following the lake drainage. Here, we apply the same methodology to characterize the spatial distribution of basal

slip and uplift during a late-season speed-up event on 2011/238. Unlike Stevens et al. (2015), we do not invert for opening along a vertical hydrofracture because no crack-normal motion is observed in the GPS positions that would indicate a hydrofracture crack opening or closing during the late-season melt event (nor is a lake present at this time). As such, we assumed that all uplift is a result of basal cavity opening and all basal slip is parallel to the local flowline direction (276–277°) for this late-season speed-up event.


A challenge in applying the NIF is that this approach requires good station coverage, is computationally expensive, and requires a relatively uniform background velocity field from which the velocity changes associated with the speed-up event can be differentiated. While the 2011/238 melt event had good station coverage (14 GPS stations, compared to 15 stations on 2011/169), many late-season speed-up events have too few stations to perform the NIF inversion. Also, the regional melt events in the mid-

to-late melt season do not always have a uniform, well-defined background velocity in period preceding the speed-up event from which the transient changes can be resolved.

Thus, we developed an alternative approach to more easily identify and characterize all transient speed-up events present during the 2011 and 2012 melt seasons, and draw comparisons between local lake drainages and regional melt events. Transient speed-

up events were identified based on having a velocity averaged across all operating GPS sensors that was ≥ 50 m/day above the average background ice velocity leading up to the speed-up event. Based on this definition, six speed-up events were identified in 2011 and seven speed-up events were identified in 2012 (Fig. 1b and c). Here, the pre-speed-up event background velocity is the average velocity recorded by each sensor over the 2–7-day period that precedes the speed-up event; to qualify as a speed-up event, the average velocity during the speed-up event must remain elevated above the pre-speed-up event background velocity for a

minimum of 24 hours. Because the background velocity changes seasonally and locally, a pre-speed-up event velocity for each speed-up event was fit individually per station. At each station we defined the velocity response ($\Delta V$) as the difference between the maximum velocity during the speed-up event and pre-speed up event velocity following Eq. (1):



$$\Delta V = V_{max} - V_{pre} \qquad\qquad (1)$$


and the normalized velocity response ($\Delta V_N$) as the ratio of maximum velocity during the speed-up event to the mean pre-speed-up event velocity following Eq. (2):

$$\Delta V_N = \frac{V_{max}}{V_{pre}} \qquad\qquad (2)$$


To determine the speed-up event duration, the start and end time was first estimated at each station. The start of the speed-up event was defined by the time at which the velocity remained elevated above the pre-speed-up event velocity for a minimum of 24 hours. The end of the speed-up event was defined as the first time when the velocity dropped below the pre-speed-up event velocity after the maximum velocity. If the velocity did not drop below the pre-speed-up event velocity the first local minimum in velocity was

used. The beginning and end times were then averaged across all stations for each speed-up event and rounded to the nearest day (Fig. 1d and e). In some instances, the temporal proximity of transient speed-up events to prior speed-up events limits the time frame over which the pre-speed-up event velocity can be determined. Ideally, the pre-speed-up event velocity would be determined from a full week of velocity observations, but we allowed pre-speed-up event velocities to be estimated from as little as 2 days.

**2.2 Modelled estimates of runoff**

To evaluate the relationship between ice velocity during each speed-up event and the input of meltwater to the subglacial hydrologic system, runoff at North Lake was estimated for 2011 and 2012 using the Regional Atmospheric Climate Model (RACMO) (Noël et al., 2015). To define the catchment basin in which North Lake resides, we used TopoToolbox (Schwanghart et al., 2014) and

the surface topography from the 10-m resolution ArcticDEM dataset (Porter et al., 2023). The magnitude of runoff supplying the subglacial system beneath North Lake, was then calculated from the average runoff across all 11-km x 11-km RACMO points within the drainage basin. This is a generalized estimate for the variable runoff that makes it to the bed directly below the lake, but could miss drainage through other moulins in the basin or overflow from other drainage basins.

Almost all speed-up events corresponded to a peak in runoff (Fig. 1b–e), and the converse is also true that almost all runoff spikes correspond to transient speed-ups. The runoff for each speed-up event was integrated over the speed-up event duration. The beginning and end times used to calculate the runoff for each speed-up event were determined using the bounds provided by the velocity response. The precise definition of the beginning and end of the temporal bounds has potential implications for the runoff magnitude; however, changing these bounds by ± 3–4 days did not affect any of the correlations we found between speed-up event

runoff and the velocity response. Using the speed-up event time bounds, we calculated the maximum, mean, and total runoff for each speed-up event.

In the case of the early-season lake drainage events, North Lake stores significant amounts of meltwater that is released rapidly into the subglacial system over a couple hours. This meltwater is not reflected in the daily RACMO runoff estimates. To account

for the volume of meltwater stored in lakes, we estimate an "effective runoff" by assuming the entire lake-basin volume is supplied to a local bed region around the lake. To prescribe this effective runoff, we use the Stevens et al. (2015) lake-basin volume estimates for the 2011 and 2012 lake drainage events. In order to directly compare these values to the RACMO runoff (measured in mm of water equivalent (w.e.) per day), we divided the lake volume by the area that the drainage distributed meltwater to the bed, inferred





from the uplift pattern associated with the lake drainage (Stevens et al., 2015). For example, during the 2011/169 lake drainage

event, the post-drainage uplift pattern was roughly circular with a radius of 2 to 3 km. Using the pre-drainage lake volume of 0.0077 km³, we calculated an effective daily runoff of 600 to 330 mm w.e for circular blisters of, respectively, 2 to 3 km. Using the lake drainage duration of 5 hours and the elevated velocity duration of 3 days, a maximum and mean effective runoff were determined. These values are roughly an order-of-magnitude greater than daily runoff in the region during this time period.


### 3 Results

Below we describe the behavior of transient speed-up events throughout the 2011 and 2012 melt seasons. We first describe the results of the NIF for the early-season lake drainage and late-season melt event in 2011. We then show how the NIF results are

consistent with the overall evolution in transient speed-up event behavior throughout the early, middle, and late melt seasons. These variations in transient speed-up event characteristics (e.g., amplitude and variability of speed-up) are then correlated to seasonal changes in runoff.

### 3.1 Comparison of velocity response for 2011 lake drainage and late-season melt events


We first used the NIF to investigate the velocity response of the 2011/238 melt event compared to the 2011/169 lake drainage previously characterized by Stevens et al. (2015). The extra basal slip, primarily expressed in the horizontal flowline displacement, was plotted relative to the onset of the speed-up event (Fig. 2). The magnitude of the 2011/238 extra basal slip (~2.0 m) is approximately 4 times greater than the 2011/169 lake drainage (~0.5 m). Similarly, the magnitude of the 2011/238 flowline vectors

(~1.0 m) is greater than the 2011/169 lake drainage (~0.5 m). Furthermore, the late-season speed-up event is characterized by a significantly more uniform flowline displacement and uplift response, as highlighted by a direct station-to-station comparison (Fig. 3). The average excess flowline displacement associated with the lake drainage event is ~0.13 m, compared to ~1.2 m for the late-season melt event. In contrast to the flowline displacements, the lake drainage event had an average uplift of ~0.6 m, which is larger than the average uplift of ~0.2 m in the late-season melt event. Thus, overall, the late-season melt event is characterized by

a larger amplitude and more uniform flowline displacement compared to the early-season 2011 lake drainage event (Fig. 3), but with a significantly smaller component of uplift. The early-season lake drainage also shows systematically greater uplift at stations near the lake that experience the greatest speed-up, while the late-season melt event shows more variable uplift across the array that does not correlate with speed-up (Fig. 3).

### 3.2 Simplified velocity analysis of 2011 and 2012 speed-up events

To extend our analysis to all 13 transient speed-up events observed in 2011 and 2012, we next applied our simplified approach for quantifying the velocity response ($\Delta V$) and the normalized velocity response ($\Delta V_N$,) and compared these results to the runoff calculated for each speed-up event. To illustrate the robustness of this approach, we first calculated the velocity response for the

2011/238 late-season speed-up event (Fig. 4). As described above, $\Delta V$ for each station was determined from the difference between the maximum velocity, $V_{max}$, and pre-speed-up event velocity, $V_{pre}$ (Eqn. 1) and the normalized velocity response, $\Delta V_N$, was determined from the ratio between $V_{max}$ and $V_{pre}$ (Eqn. 2). The average pre-speed-up event velocity for 2011/238 is ~65 m/yr, and the average maximum velocity is ~200 m/yr. For the 2011/238 melt event, $\Delta V$ ranges from 98.5–151.9 m/yr (Fig. 4), with an





average $\overline{\Delta V}$ of 135.5 m/yr and a standard deviation of ~16.2 m/yr. The normalized velocity response, $\Delta V_N$, ranges from ~2.8–3.4,

with an average $\overline{\Delta V_N}$ of 3.1 and standard deviation of ~0.2. The uniformity and relatively large magnitude of $\Delta V$ and $\Delta V_N$ is

consistent with the flowline displacements determined by the NIF (Fig. 3). Integrating the average $\overline{\Delta V}$ of the 2011/238 melt event

over its 8-day duration, gives an estimated displacement of 3.0 m or 3.6% of the region's annual displacement.

For comparison, we evaluated the velocity response of the 2011/169 lake drainage event (Fig. 5). This lake drainage event has

been studied in detail by Stevens et al. (2015), and thus we used their definition of 2011/168.85 for the pre-speed-up event start

date. We note that the 2011/169 lake drainage event occurs shortly after the onset of the summer speed-up; however, this increase

over the pre-season winter velocity is not included as part of the pre-speed-up event velocity estimate because it occurs before the

relatively stable precursor phase identified by Stevens et al. (2015). Furthermore, because the velocity at some stations does not

decrease to pre-speed-up event velocities in a timescale to accurately define the end of the velocity transient (e.g., NL09; Fig. 5),

a date of 2011/172 was utilized as the end time based on the local minima in velocity at the stations following $V_{max}$. Because the

maximum velocity takes place on ~2011/170, the end time of the speed-up event does not alter the velocity response calculations,

which are reliant only on the pre-speed-up event velocity and the maximum velocity. Across all stations, the average pre-speed-up

event velocity for 2011/169 is ~158.3 m/yr and the average maximum velocity is ~236.6 m/yr. The $\Delta V$ for this lake drainage event

ranges from 12.6–175.6 m/yr with an average velocity response $\overline{\Delta V}$ of ~78.3 m/yr and a standard deviation of ~56.5 m/yr across

the array. The normalized velocity response, $\Delta V_N$, ranges from 1.1 to 2.1 with an array-average $\overline{\Delta V_N}$ of ~1.5 and a standard

deviation of ~0.3. The average velocity across the array from 2011/168–172 is ~192.8 m/yr. Integrating the average $\overline{\Delta V}$ of the lake

drainage, 78.3 m/yr, over the lake drainage event duration provides an estimated displacement of 0.86 m or 1% of the region's

annual displacement. These results are consistent with the NIF findings, which similarly show a variable, muted velocity response

(Fig. 3).


Compared to the 2011/238 late-season melt event, the variability of the response as inferred by the standard deviation in $\Delta V$ and

$\overline{\Delta V_N}$ across the array is notably greater for the lake drainage event. The larger $\Delta V$ and duration of the 2011/238 late-season melt

event compared to the 2011/169 lake drainage, results in a greater effect on annual displacement. The annual displacement

discrepancy is highlighted when considering the average $\overline{\Delta V}$ integrated over the speed-up event duration, in which the lake drainage

accounts for only 0.86 m or 1% and the late-season melt event for 3.0 m or 3.6% of the region's annual displacement. In addition,

the $\overline{\Delta V_N}$ is significantly greater for the 2011/238 melt event than it is for the lake drainage event, indicating that velocities during

the late-season transient speed-up are more elevated above the pre-speed-up event velocities compared to the early-season lake

drainage.

**3.3 No correlation between runoff and transient speed-up response**

Following this same approach, the relationship between the velocity response and runoff was explored for all 13 transient speed-

up events in 2011 and 2012. In general, $V_{pre}$ for each speed-up event is related to DOY (Fig. S1) reflecting a steady decline in

background velocity from the early-season peak to late-season minimum (Stevens et al., 2016). For each station, the maximum

runoff, mean runoff, and integrated runoff were compared to $\Delta V$ (Fig. 6a–c) and $\Delta V_N$ (Fig. 6d–f). The maximum for $\Delta V$ reaches

a value of ~200 m/yr in both 2011 and 2012, while $\overline{\Delta V}$ for each speed-up event does not exceed ~150 m/yr. As described above,

for the North Lake drainage events on 2011/169 and 2012/162, the previously estimated lake volume was used instead of the





RACMO modelled runoff, leading to an "effective runoff" that is much greater compared to the other speed-up events. Overall, we find that while transient speed-ups coincide with melt events as shown in Figure 1, there are no easily identifiable systematic

trends between the magnitude of the velocity response and the speed-up event runoff characteristics (Fig. 6). We will return to this point in the Discussion Section where we discuss the importance of the evolving subglacial hydrologic conditions on modulating the transient ice response to an individual melt event (e.g., Schoof, 2010).

### 3.4 Magnitude and variability of speed-up throughout the melt season


We also examined the temporal evolution of $\overline{\Delta V}$ and $\overline{\Delta V_N}$ throughout the melt season, both for all speed-up events collectively, and also for the middle and late-season regional melt events that do not coincide with local lake drainage events (Fig. 7). Overall, two main trends are observed in the data. First, there is a general increase in the magnitude of the velocity response, which is most clearly reflected in the normalized velocity response $\overline{\Delta V_N}$ (Fig. 7a and b). Specifically, $\overline{\Delta V_N}$ increases through time with an R-

squared of 0.44 and a *p*-value of 0.01 (black, Fig. 7b). Removing the local lake drainage events from this analysis, a similar trend can be observed among the regional melt events, which have an R-squared of 0.49 and a *p*-value of 0.02 (magenta, Fig. 7b). Second, the variability in the velocity response, as determined by the standard deviation in $\Delta V$ and $\Delta V_N$, decreases throughout the melt season both including and excluding the local lake drainage events (Fig. 7c and d). This trend is consistent with the differences initially seen in the NIF results for the 2011 lake drainage (2011/169) and late-season melt event (2011/238); however, as shown

by the analysis of the regional melt events alone this trend extends beyond just reflecting higher variability associated with the two, known North Lake drainages. In particular, after DOY 200 in both years, there is significantly less variability in $\Delta V$ (Fig. 7c). The R-squared value of the linear fit to the standard deviation $\Delta V$ versus DOY is 0.36 with a *p*-value of 0.03 when considering all speed-up events, and has an R-squared of 0.25 and *p*-value of 0.15 when considering the regional melt events only, indicating little in the way of a trend after lake drainages have ceased. Similarly, there is little in the way of a trend with time for the standard

deviation in $\Delta V_N$ when considering all speed-up events (R-squared = 0.12; *p*-value = 0.25; Fig. 7d). We also find that because $V_{pre}$ is correlated to DOY (Fig. S1), similar trends are found when comparing the velocity response parameters to $V_{pre}$ (Fig. S2). Finally, we note that the largest variability is associated with the 2012/180 speed-up event, which does not correspond to a North Lake drainage. We infer this speed-up event is associated with the drainage of a neighboring lake and thus classify it as a local "flood" event; it will be discussed further in Section 4.2 below.


### 4 Discussion

The sliding behavior of the Greenland Ice Sheet and its relationship to summer melt are linked through the evolution of the

subglacial hydrologic system and its influence on basal sliding (Schoof, 2010; Hewitt, 2012; Chandler et al., 2013; Andrews et al., 2014; Flowers, 2015; Joughin et al., 2013). Recently, Lai et al. (2021) used observations of uplift relaxation following lake drainage events occurring at different times of the melt season to probe this relationship, finding that, in general, mid-season lake drainage events are characterized by shorter duration speed-up transients compared to those that occur earlier in the melt season. They interpreted this finding to reflect the increasing transmissivity of the basal hydrologic system as a more channelized system

develops throughout the melt season. However, because lake drainage events typically do not occur late in the melt season, Lai et al. (2021) were unable to probe the full, seasonal evolution of the basal hydrologic system. In particular, they did not resolve the period late in the season when models predict that decreasing runoff input is unable to outpace creep closure of channels (Schoof,



2010). Further, Andrews et al. (2014) used *in situ* observations in the mid-ablation zone to show that channelization could account for decreasing velocities in the early melt season, but not in the late melt season. Instead, they proposed that the formation of flow pathways and/or connectivity between unchannelized regions of the bed drive the late-season increase in drainage-system efficiency (Andrews et al., 2014). Hoffman et al. (2016) further argued that the dewatering of weakly connected basal cavities is necessary to describe late-season subglacial conditions. They proposed that in the late-season, while the majority of subglacial channels have closed, these low-water-pressure, poorly connected cavities persist and likely drive the observed decrease in ice-sheet velocities until they are able to refill by basal melting, returning the system to its winter velocity (Hoffman et al., 2016). Here we discuss our results in the context of the subglacial conditions throughout the melt season and the velocity response associated with distal lake drainages.

**4.1 Hypothesized subglacial conditions of early-season lake drainages vs. late-season melt events**

The transient velocity response to runoff events extend the Lai et al. (2021) lake drainage dataset into the late melt season, at a time when background ice velocities (~65 m/yr) have dropped below the local "background" winter velocities (74–76 m/yr in 2011–2012; Stevens et al., 2016). The NIF analysis of the 2011/238 late-season melt event shows clear differences from early-season supraglacial lake drainage events. Specifically, the NIF describes the 2011/238 melt event as having a more-uniform, longer duration, and higher amplitude speed-up in the flowline direction, but with a significantly smaller maximum uplift as compared to the 2011/169 lake drainage event earlier that same year (Fig. 3). An explanation for the lack of late-season uplift is likely the smaller volume of meltwater delivered to the bed over a longer period of time; the runoff associated with the 8-day 2011/238 melt event is roughly an order-of-magnitude less than the "effective runoff" of the 4-day lake drainage event. This difference is intriguing, given that the larger velocity response would conventionally imply greater decoupling of the ice sheet from its bed, but in this case without the associated uplift typically observed during lake drainage events. These results argue against a strongly channelized basal meltwater system at 2011/238 since a well-developed system would be expected to quickly evacuate the runoff (consistent with the lack of an uplift signal), but simultaneously reduce the magnitude and duration of the sliding transient. Therefore, it is reasonable to assume that any channelized network formed during the melt season has closed substantially by 2011/238—implying that channels may not be the primary reason that the late-season background velocity remains below the winter velocity.

Similarly, our analysis of the additional local and regional velocity transients in 2011 and 2012 supports the hypothesis that the state of the subglacial drainage system influences the velocity response to a greater extent than runoff magnitude. While we stress that all velocity transients are linked to melt events, our results show that the transient velocity response above the background rate (estimated by $V_{pre}$) is poorly correlated to runoff magnitude. We find no correlation between $\Delta V$ or $\Delta V_N$ and the maximum, mean, or total integrated flux during a speed-up event (Fig. 5). However, $\Delta V$ and $\Delta V_N$ do increase late in the melt season, while their variability decreases (Fig. 7). Importantly, these relationships hold not just when comparing late-season regional melt events to early-season lake drainages, but also when comparing the late-season melt events to regional melt events that occur earlier in the same melt season. This suggests the state of the subglacial system when the water reaches the bed drives these trends, rather than the style of melt water delivery to the bed.

In 2012, many of the regional melt events between DOY ~170 to ~250 occur while background velocities are decreasing (Fig. S1). If considering the background velocities in isolation this trend could be interpreted as evidence for channelized conditions until




the minimum velocity at DOY 250. However, the transient speed-ups show a trend toward greater amplitude velocity responses and lower variability over this same period, suggesting the channels may be closing even as the background velocities are still

decreasing. Further, it does not appear that the input of melt during the mid- to late-season melt events significantly modify the drainage system, because the pre- and post-transient velocities remain similar for each speed-up event. Thus, the transient speed-ups provide an effective proxy to examine the subglacial conditions as the melt pulses temporarily overwhelm drainage system but do not reset it.

One possible interpretation of these results is that during the early-season, the ice sheet is still largely coupled to the glacial bed across the region. When a lake drainage occurs, the ice sheet decouples from the bed directly below the lake, but frictional resistance from the surrounding regions buffers the overall velocity response. This is consistent with the heterogeneous spatial distribution of speed-up associated with lake drainages observed with more than one GPS sensor (e.g., Doyle et al., 2013; Stevens et al., 2015). For North Lake, stations NL04, FL03, NL07, NL08, and NL10 are closest to the lake and show the largest lake drainage velocity

response, as well as the largest uplift signal (Stevens et al., 2015; Figs. 2 and 3). These findings are also consistent with observations of spatially variable ice velocities on ~weekly timescales across the region (Joughin et al., 2013). Specifically, Joughin et al. (2013) argued that higher velocities correspond to meltwater pooling in a basal topographic trough running from NE to SW through this region (Joughin et al., 2013).

In contrast, we speculate that the late-season melt event reflects a much broader and more uniform input of melt to the bed, possibly into a cavity-dominated system, influencing a larger area and producing the observed higher amplitude, more uniform sliding response. Past research has found that after the drainage of supraglacial lakes, moulins beneath lake basins remain open throughout the remainder of the melt season (Flowers, 2015). Thus, because most supraglacial lake drainages occur in the early-to-mid melt season, by the late-season there should be many open moulins available to provide direct surface-to-bed conduits for surface runoff

(e.g., Krawczynski et al., 2009). Assuming these moulins remain open into the late-season, they will provide a pervasive and relatively uniform network of access points to the bed (Joughin et al.., 2013; Yang et al., 2016). The dissociation of uplift with the late-season melt event could thus reflect the smaller magnitude runoff and more widespread input of melt to a cavity-dominated subglacial system. Overall, this points to a greater sensitivity of ice velocity to late-season melt input, consistent with observations by Doyle et al. (2015). However, as noted by Ing et al. (2024), the 2011/238 melt event only contributes a small amount (~3.4%)

to the annual ice motion at North Lake, implying that unless such late-season runoff events become more frequent, they do not constitute a major fraction of the ice-sheet motion in this region.

The combination of a relatively large, uniform velocity response, but small uplift signal in the late-season melt events indicates that meltwater is distributed in such a way that allows large, homogenous ice accelerations. Further, the onset of these speed-up

events provides preliminary information on the timing of the evolution of the subglacial hydrologic system away from mid-summer conditions. We present a conceptual model for the evolution of the subglacial hydrologic system and its relation to ice-flow dynamics in Figure 8. Early in the season, lake drainages result in large uplift, but smaller amplitudes relative to the pre-speed-up event horizontal sliding transients. This response reflects the presence of the water filled cavities without an established channel networks to efficiently transport melt, producing high water pressures at the bed. A blister of water forms beneath the lake basin,

resulting in uplift directly below the lake and a non-uniform velocity transient across the array, with stations closest to the lake having the most pronounced velocity response (Fig. 8a). The formation of this blister is enabled due to the relatively low transmissivity of the unchannelized hydrologic system in the early-season (e.g., Lai et al., 2021). The horizontal-velocity increases



associated with early-season lake drainages are likely muted by the regions around the lake that remain coupled to the bed; this strong coupling persists when there is a lack of additional surface-to-bed meltwater conduits. As the melt season progresses more drainages occur, creating or reopening moulins, and supplying large volumes of water to the bed. This high rate of melt input produces high flux, lower pressure channels, which evacuate meltwater from other areas of the bed, increasing frictional coupling, and leading to the initial slow-down of the ice sheet (e.g., Schoof, 2010; Hewitt, 2013; Fig. 8b). Finally, late in the melt season, decreased runoff causes channel closing by viscous creep outpacing opening by melt at time-scales of days (Bartholomous et al., 2011). As a result, the subglacial system becomes more inefficient. However, moulins that opened throughout the melt season likely remain open, resulting in a pervasive network of surface-to-bed conduits. Taken together these two effects allow smaller magnitude regional melt events to decouple the bed over much broader areas compared to earlier in the season, producing larger, and more uniform, velocity transients in the late-season (Fig. 8c).

Building on the ideas of Andrews et al. (2014) and Hoffman et al. (2016) late-season melt events may supply sufficient meltwater to the bed to temporarily fill dewatered cavities, increasing short-term velocities. If flow pathways between cavities in the late melt season have crept closed due to increased effective pressure, widespread melt inputs may be able to temporarily overwhelm the subglacial system. The long durations of the late-season melt events on 2011/238 and 2012/228 (7 and 9 days, respectively) indicates decreased transmissivity and the inability of melt inputs to re-establish cavity connectivity.

**4.2 Variable sliding response during lake drainage events outside of the GPS array**

It has been observed that velocity transients can also result from flood events caused by nearby lake drainages, with melt flowing to lower elevations, as dictated by the basal topography (Andrews et al., 2018; Mejia et al., 2021; Stevens et al., 2022). The mid-season speed-up event on 2012/180 shows the greatest variability in sliding response of all speed-up events analyzed in 2011 and 2012, including the North Lake drainage events (Fig. 7c and d). The $\overline{V_{pre}}$ for 2012/180 was ~156 m/yr and the $\overline{V_{max}}$ was ~272 m/yr (Fig. 9). The $\Delta V_N$ ranged from 1.2–2.6, with an $\overline{\Delta V_N}$ of 1.7 and standard deviation of 0.5 (Fig. 9). However, the GPS stations that showed the greatest $\Delta V$ and $\Delta V_N$ during this speed-up event (NL11, NL12, NL13; Fig. 9) differed from the GPS stations most responsive during North Lake drainages (NL7, NL8, NL10; Fig. 5).

To assess what caused the high variability of this speed-up event, we analyzed available Landsat-7 satellite images before and after the speed-up event. The Landsat images show a local lake drainage ~8 km to the northeast of the North Lake basin occurred sometime between 2012/171 and 2012/178 (Fig. 10). Further, the spatial pattern of the velocity response shows the largest $\Delta V$ in the south and smallest $\Delta V$ in the north of the GPS array. This pattern is spatially correlated with the basal topography in the region (Morlighem et al., 2017), with the largest velocities coinciding with the lowest bed elevations. We interpret this to reflect that meltwater from the supraglacial lake drainage to the northeast has been preferentially transported down the hydraulic potential gradient (Chu et al., 2016), pooling in the bedrock basin to the south of North Lake. These results are consistent with Joughin et al. (2013), who described a region of elevated velocities that occurs seasonally and is aligned with the bedrock trough that extends northeast-southwest beneath North Lake. These results strongly suggest that bedrock topography influences local patterns of meltwater flow and ice-bed coupling. Moreover, the rate of subglacial flow must be no greater than ~10 km/day (assuming the lake drainage event occurred immediately before the 2012/178 Landsat image) and no less than ~1.25 km/day (assuming the lake drainage event occurred immediately after the 2012/171 Landsat image). For comparison, Hoffman et al. (2016) observed down-glacier flood propagation speeds ~26 km/day following a supraglacial lake drainage in 2011 in west Greenland. These observations





also point to the potential for hydro-fracture event triggering between adjacent lakes associated with stress coupling due to either ice speed-up or uplift associated with focused basal meltwater transport (Stevens et al., 2024).


In the context of the full melt season, the 2012/180 speed-up event is quite short in duration (~1 day) and the background velocities dropped rapidly after the speed-up event (Fig. 8e), indicating a change in subglacial conditions towards a more efficient system. Lake drainages are often indicative of the onset of channelization (Andrews et al., 2018); however, some lake drainage events have been shown to slow ice-sheet velocities by dewatering of subglacial cavities without enlarging subglacial channels (Mejia et al.,

2021). Additional observations of moulin water levels or focused subglacial hydrology modelling would be required to determine if this speed-up event is evidence of a cavity- or channel-related slow down.

Finally, we explored whether spatial patterns also exist between the surface-velocity response and sensor location across the North Lake region over the entirety of the melt season. For each speed-up event in 2011, we calculated the difference in the speed-up

($\Delta V$) at each station relative to the speed-up averaged across the array ($\overline{\Delta V}$). Overall, there are slightly smaller speed-ups relative to the array average in the northern half of the array, compared to larger speed-ups in the southern half (Fig. 11). An equivalent spatial distribution of speed-up was observed over the 2012 melt season (not shown), beyond the 2012/180 speed-up event (Fig. 10). These results further point toward the importance of basal topography in controlling patterns basal hydrology and ice-bed coupling on regional spatial scales.


## 5 Conclusions

This study builds upon a growing body of knowledge of the evolution of subglacial bed conditions and their effect on ice-sheet

acceleration. Our findings provide preliminary insights into the structure and temporal evolution of the subglacial hydrologic system beneath the ablation zone on western margin of the Greenland Ice Sheet. We find enhanced ice-flow sensitivity to melt input in the form of longer, more uniform, velocity responses during late-season, runoff-induced, speed-up events compared to those of early- to mid-season lake drainage or regional melt events. However, the uplift signal associated with these late-season melt events is small and spatially heterogeneous, in contrast to lake drainage events in which meltwater is focused to a single

location at the bed, producing pronounced, but spatially coherent uplift in a localized region. We interpret our results to imply that in the late melt season, most subglacial channels and/or connective flow pathways between cavities have substantially closed, sharply lowering basal transmissivity. At the same time, moulins formed throughout the melt season, likely remain open, allowing for pervasive and widely distributed surface-to-bed pathways for meltwater to reach the bed. The culmination of these factors results in late-season melt events that rapidly overwhelm the subglacial system and decreasing frictional

coupling at the bed over larger spatial scales than lake drainages or regional melt events earlier in the season. Due to their extended duration and amplitude, these late-season melt events accommodate a larger fraction of the annual ice motion compared to lake drainage at North Lake; however, their net influence on ice sheet motion remains small (2–3% of annual displacement). Finally, we document that migration of meltwater pulses from lake drainages can influence sliding behavior over distances of ~10 km, and that migration of these pulses appears to follow local bedrock topographic lows.


## Data Availability



GPS data are archived at the GAGE Facility operated by the EarthScope Consortium

(https://www.unavco.org/data/doi/10.7283/T5222SJK; Das et al., 2018).

Sentinel-2 Imagery from the European Space Agency were accessed via Esri ArcGIS Pro

(https://sentinel.arcgis.com/arcgis/rest/services/Sentinel2/ImageServer).


**Code Availability**

A version of the Network Inversion Filter (NIF) code for North Lake drainages is archived in a Zenodo repository

(https://doi.org/10.5281/zenodo.10650188; Stevens et al., 2024).


**Author Contributions**

G. Gjerde performed the data analysis with oversight from M. D. Behn and L.A. Stevens. NIF code was developed by L.A. Stevens.
S.B. Das, I. Joughin, and M.D. Behn conceived the project and collected the data with L.A. Stevens. G. Gjerde prepared the manuscript with contributions from all co-authors.

**Acknowledgements**


Support was provided by the National Science Foundation's Office of Polar Programs (NSF-OPP) and National Aeronautics and Space Administration's (NASA's) Cryospheric Sciences Program through ARC-0520077, ARC-1023364, and NNX10AI30G to S.B. Das and M.D. Behn; OPP-1838410 to M.D. Behn; and ARC-0520382, ARC-1023382, and NNX10AI33G to I. Joughin. G. Gjerde would also like to the Boston College Undergraduate Research Fellows Program. L.A. Stevens acknowledges funding from
the John Fell Oxford University Press Fund and the UK Natural Environment Research Council (NE/Y002369/1). Logistical and instrumental support was provided by UNAVCO and CH2MHILL Polar Field Services. We thank Yi Ming for suggestions on an early version of this manuscript.

**Competing Interests**

The authors declare they have no competing interests.

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



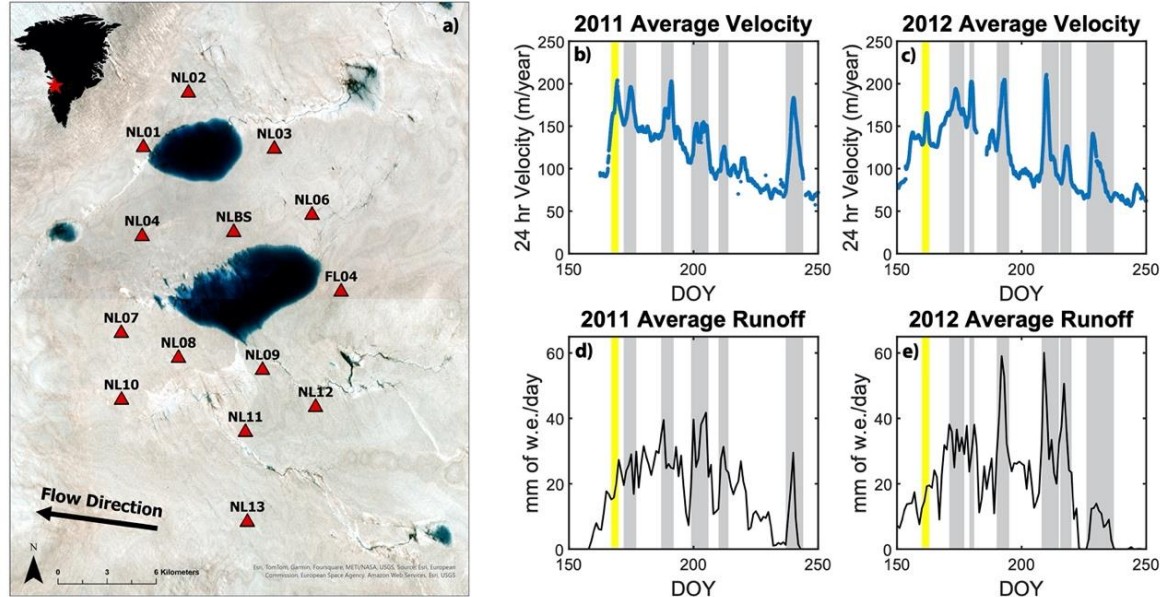

**Figure 1. (a)** Sentinel 2 satellite imagery of study area July 2018. Sentinel 2 processed by Esri. Inset shows location of study area (red star). Red triangles in panel **(a)** represent GPS sensor locations around North Lake (at center). Annual ice flow direction is indicated. **(b–c)** Smoothed 24-hour velocity across GPS array and **(d–e)** runoff estimates based on RACMO for 2011 and 2012, respectively. Grey bars denote time periods of the speed-up events, which were used to calculate event runoff. Yellow bars show North Lake drainage events via hydro-fracture in which there is additional input of 330 to 660 mm w.e. as described in the text.



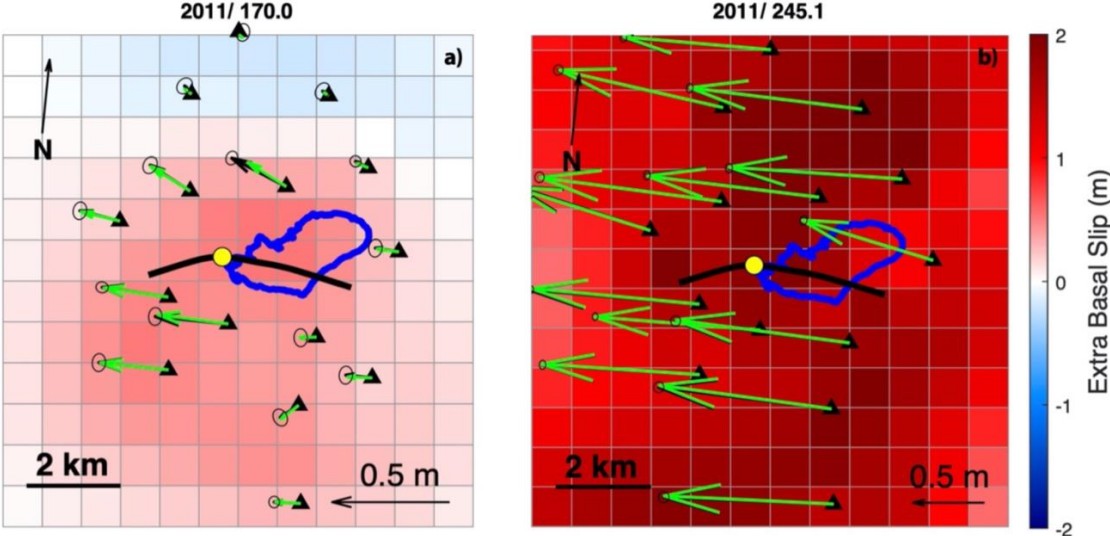

**Figure 2.** Maximum event basal slip for **(a)** 2011/169 lake drainage on DOY 170.0, and **(b)** 2011/238 on DOY 245.1. Moulin location is denoted with yellow dot, hydro-fracture crack is shown by thick black line, and North Lake basin is outlined in blue. GPS sensor locations are shown by black triangles with the black arrows representing the GPS displacement and green arrows show NIF displacement. Dark red colors (+) indicate greater basal slip than background and blue colors indicate (-) less than background values.



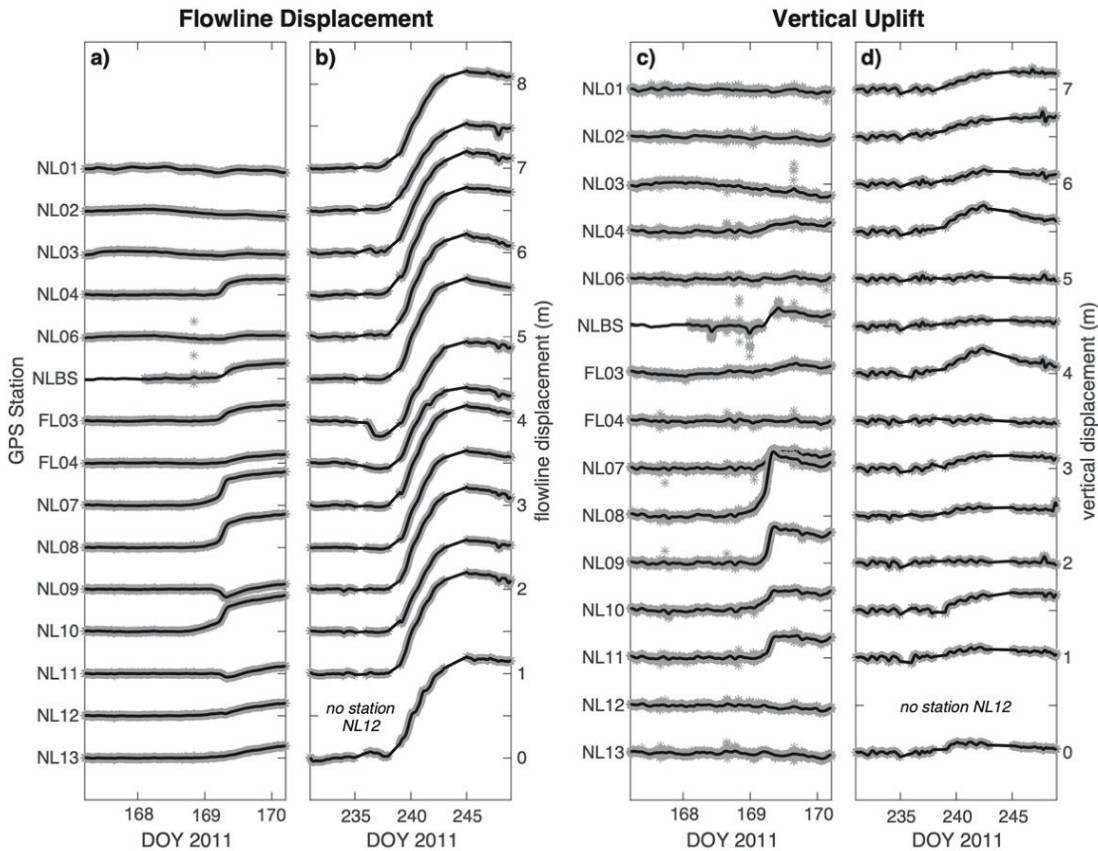

**Figure 3. (a–b)** Flowline displacement and **(c–d)** uplift displacement during the 2011/169 lake drainage and 2011/238 late–season events for all recording GPS stations (station name on left *y*-axis). The black lines represent the fit of the NIF filter and the grey dots show the recorded GPS observations.





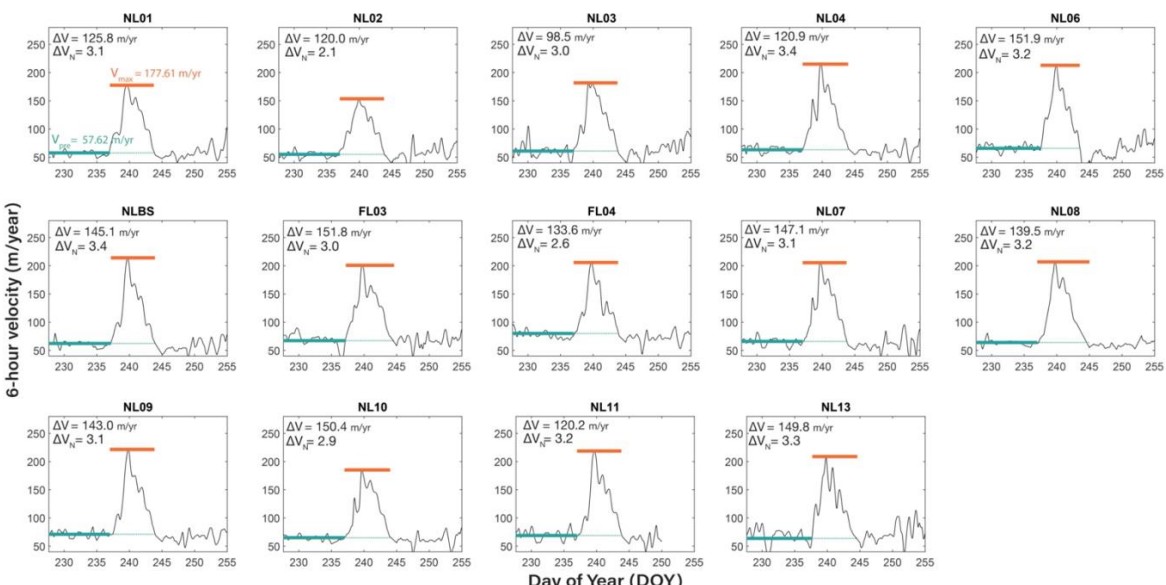


**Figure 4.** Velocity time series at all stations for the 2011/238 late-season speed-up event. Blue lines show the pre-event time period and average velocity ($V_{pre}$). Orange lines show the maximum velocity ($V_{max}$) throughout the event. Speed-up magnitudes of $\Delta V$ ($V_{max} - V_{pre}$) and $\Delta V_N$ ($V_{max}/V_{pre}$) are given in upper left corner of each panel.

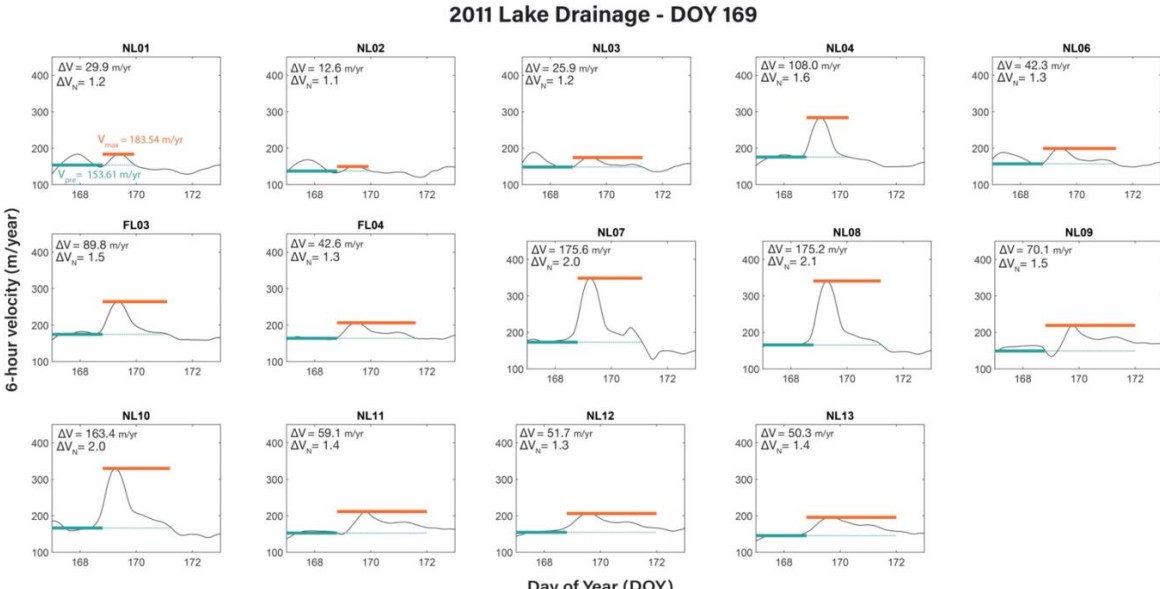

**Figure 5.** Velocity time series at all stations for the 2011/169 lake drainage event. Blue lines show the pre-event time period and average velocity ($V_{pre}$). Orange lines show the maximum velocity ($V_{max}$) throughout the event. Speed-up magnitudes of $\Delta V$ ($V_{max} - V_{pre}$) and $\Delta V_N$ ($V_{max}/V_{pre}$) are given in the upper left corner of each panel.






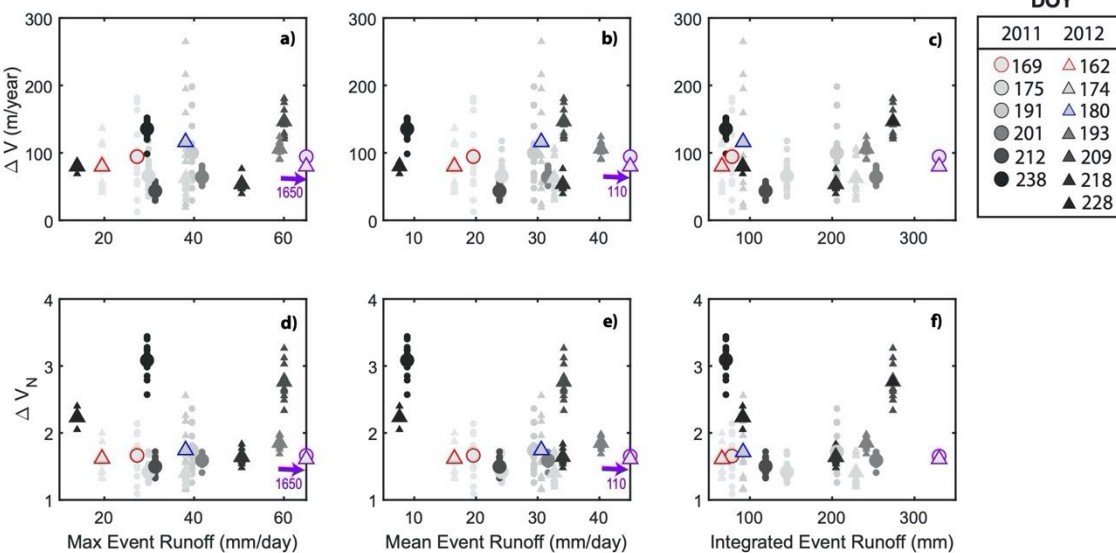

**Figure 6.** Velocity response (ΔV and $\Delta V_N$) as a function of **(a,d)** maximum event runoff, **(b,e)** mean event runoff, and **(c,f)** integrated event runoff. Circles show events in 2011; triangles show events in 2012. Individual symbols show ΔV **(a–c)** and $\Delta V_N$ **(d–f)** for each individual GPS station. Larger symbols show the $\overline{\Delta V}$ **(a–c)** and $\overline{\Delta V_N}$ **(d–f)** across all stations for each event. Colors darken chronologically with the lightest grey colors indicating events early in the melt season and the darkest black colors representing events late in the melt season. Note the lack of trend. The red-outlined symbols highlight the lake drainage events. The blue-outlined symbols highlight the 2012/180 nearby lake drainage event. The purple-outlined symbols show ΔV and $\Delta V_N$ for the lake drainage events using "effective runoff." In panels **(a–b)** and **(d–e)**, the maximum and mean "effective runoff" significantly exceed the x-axis, with the associated runoff value annotated.





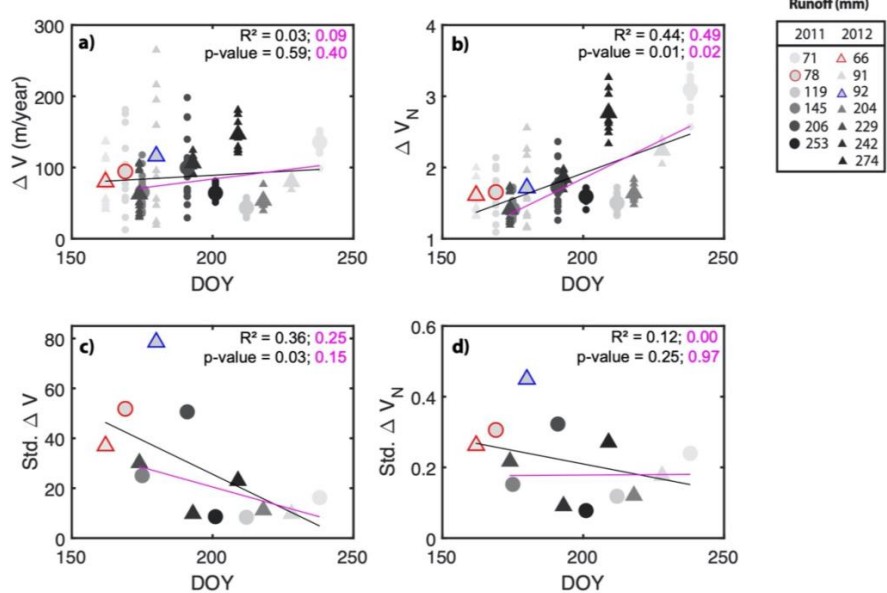

**Figure 7.** Velocity response **(a)** $\Delta V$ and **(b)** $\Delta V_N$, as a function of day of year (DOY). Circles and triangles show events in 2011 and 2012, respectively. Individual, smaller symbols in panels **(a)** and **(b)** show individual GPS stations. Larger symbols show the average value across all stations per event. Colors darken with integrated event runoff (mm), with the lightest grey color indicating less runoff and the darkest black color representing the greatest runoff. The red-outlined symbols show the North Lake drainage events. The blue-outline symbols show the neighboring lake drainage event on 2012/180. Standard deviation of **(c)** $\Delta V$ and **(d)** $\Delta V_N$, as a function of DOY. Linear fits of all events (black lines) and regional events only (magenta lines) are displayed for all panels, with associated R-squared and p-values shown in panel corners.

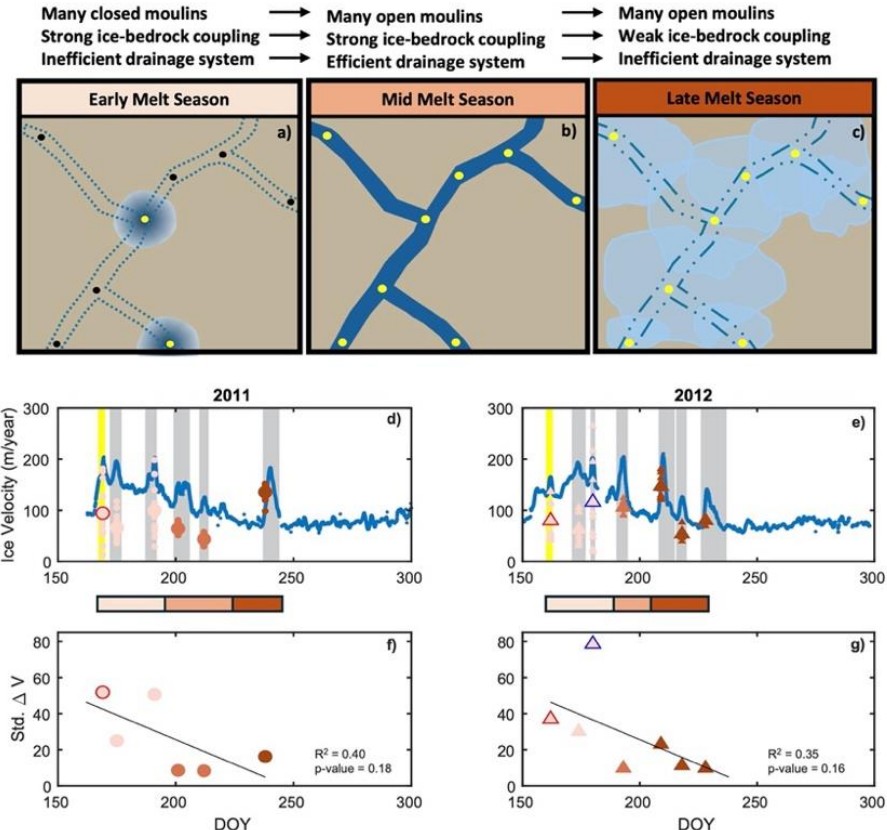

**Figure 8.** Conceptual model for subglacial conditions and ice response to melt input during **(a)** early-season lake drainage events, **(b)** mid melt season, and **(c)** late melt season. Yellow solid circles represent open moulins and black solid circles represent closed moulins. Lighter blue colors indicate lesser volumes of meltwater at the bed **(a, c)**. Dark blue colors indicate greater volumes of meltwater at the bed **(a–b)**. The circles and triangles show the ΔV (m/yr) of each speed-up in **(d)** 2011 and **(e)** 2012, respectively. Individual symbols in panels **(c)** and **(d)** represent GPS sensors, and the larger symbols are the array-average for each speed-up event. The red-outlined symbols are lake drainages and the blue-outlined symbol is a local lake drainage. The blue line shows the 24-hour average velocity across the GPS array. Light red colors indicate early-season events, medium red colors indicate middle season melt events, and dark red colors indicate late-season melt events **(d–g)**. The circles and triangles show the average standard deviation of ΔV of each speed-up event in **(f)** 2011 and **(g)** 2012, respectively.

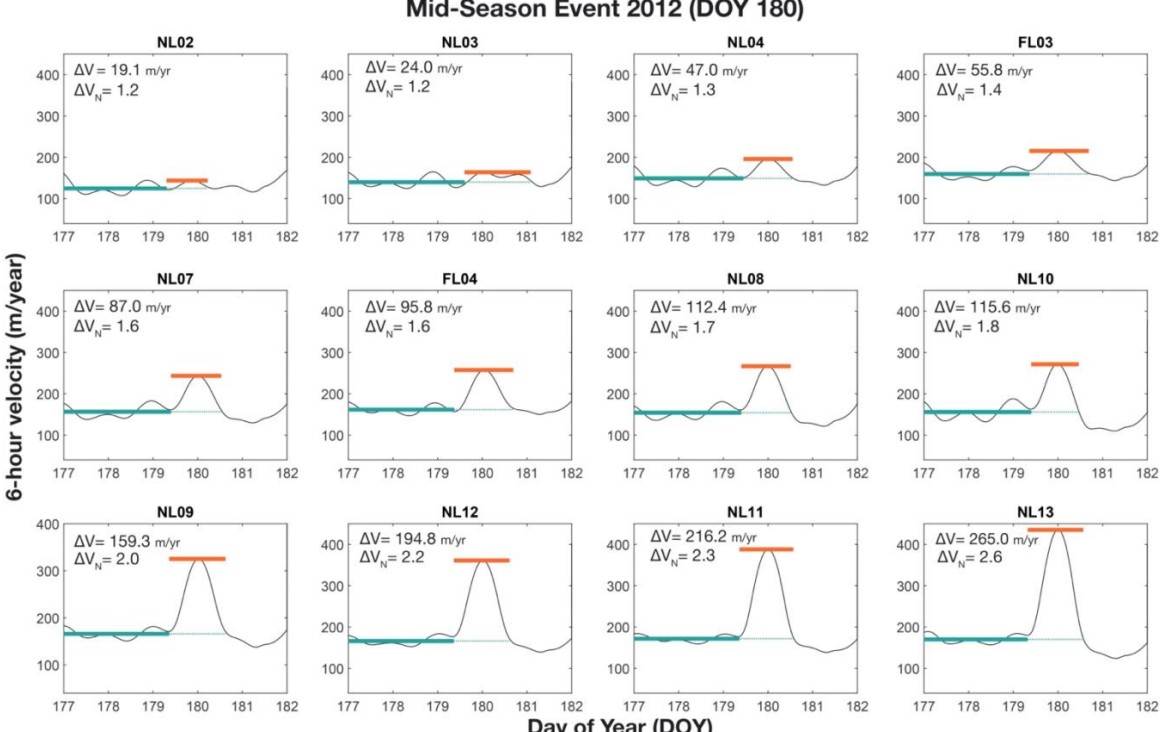


**Figure 9.** Velocity time series at all stations for the 2012/180 speed-up event. Blue lines denote the pre-event time period and average pre-event velocity ($V_{pre}$). Orange lines show the maximum velocity ($V_{max}$) throughout the event. Speed-up magnitudes of $\Delta V$ and $\Delta V_N$ are shown in in upper right corner of each panel. Stations are ordered by increasing magnitude of velocity response.



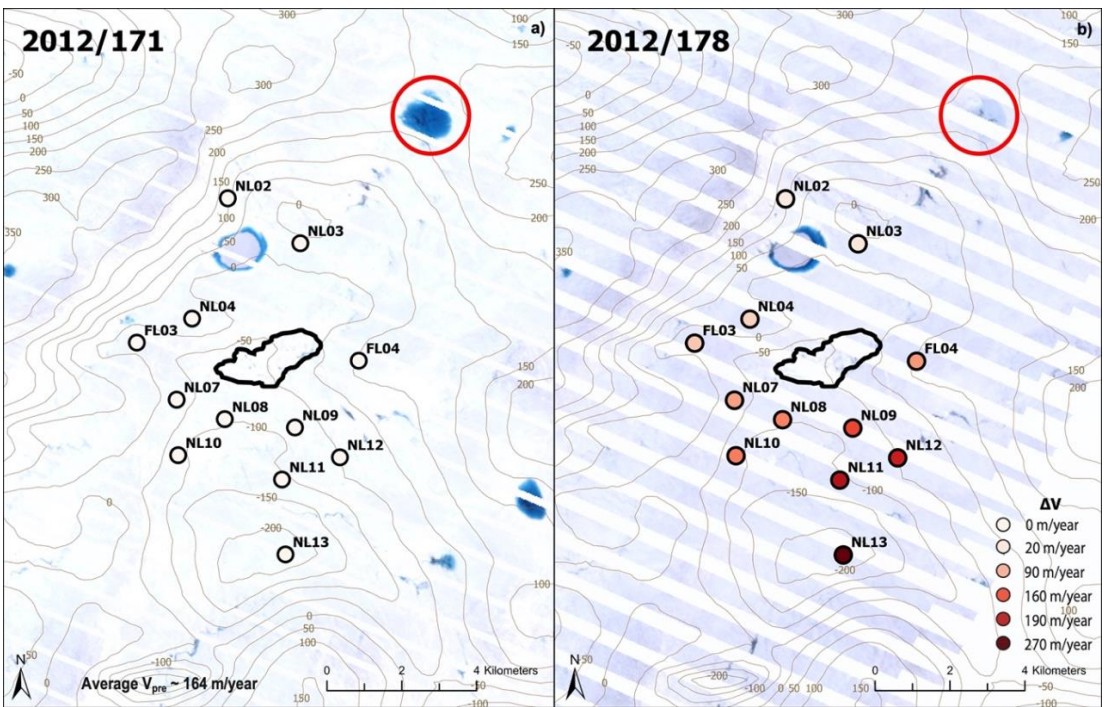


**Figure 10.** Landsat-7 satellite map of supraglacial lake catchment basin prior to 2012/180 transient speed-up event. Landsat-7 image processed by the ESA. Circles show GPS sensor locations. White colors **(a)** on 2012/171 denote small ΔV (i.e., velocities equivalent to background velocities). Redder colors on **(b)** 2012/178 plot large ΔV (i.e., velocity magnitudes above background

velocities). Brown contours show basal topography from BedMachine v3 (Morlighem et al., 2017). Note the larger velocities recorded by the southern stations tend to correspond to the lowest bed elevations.



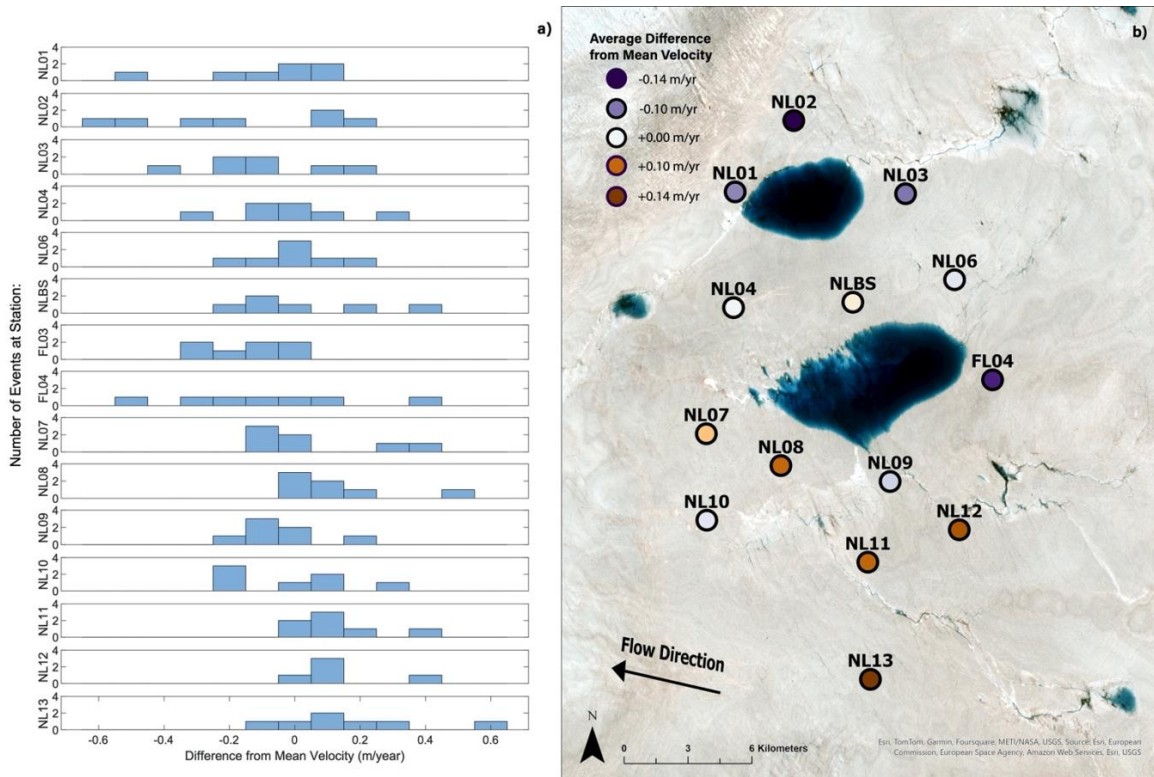


**Figure 11. (a)** Deviation of ΔV (m/yr) from the array mean ($\overline{\Delta V}_{array}$) at each station for each of the 7 transient speed-up events in 2011. **(b)** Average speed-up at each GPS sensor location for all events. Circles represent individual station locations: red colors show greater than $\overline{\Delta V}_{array}$ and purple colors show lesser than $\overline{\Delta V}_{array}$. Note the similarity in the spatial pattern of speed-up compared to the 2012/180 neighboring lake drainage event.
