# Peer review of "Seasonal drainage-system evolution beneath the Greenland Ice Sheet inferred from transient speed-up events"

_EGUsphere, 2024_

## Author Response (AR1)

Black Text – original comment from reviewer (line #s correspond to previous draft)

Red Text – submitted response to review comment (line #s correspond to previous draft)

Blue Text – edits made to revised version of manuscript (line #s correspond to revised version of manuscript with tracked changes)

**Reviewer 1**

- 1. Regarding the 2011/238 event and the calculated runoff, have the authors look into if there are other factors to consider in their runoff estimate such as precipitation (rainfall) that occurred over this period to contribute to the larger velocity response?
  - Yes, it is likely that a precipitation event occurred around the 2011/238 event due to precipitation observations (Doyle et al., 2015; Loeb et al., 2022). A week of warm, wet cyclonic weather was observed in early September 2011, resulting in enhanced surface melt and rainfall (Doyle et al., 2015). However, Doyle et al. (2015) found the magnitude of runoff and precipitation to still be less than that during the mid-melt season. As a result, Doyle et al. highlighted the contribution of an inefficient subglacial drainage system to the acceleration of ice flow during the late season. We will clarify the role of precipitation in the 2011/238 event and add these key points and references to a revised version of the manuscript.
  - Added text on lines 428-432
  - Added Loeb et al., 2022 to references
- 2. L70: The following citation should be added in describing the limited role of conduit growth during lake drainages: Dow, C. F., Kulessa, B., Rutt, I. C., Tsai, V. C., Pimentel, S., Doyle, S. H., As, D. Van, Lindbäck, K., Pettersson, R., Jones, G. a., & Hubbard, A. L. (2015). Modeling of subglacial hydrological development following rapid supraglacial lake drainage. Journal of Geophysical Research: Earth Surface, 120, 1127–1147. https://doi.org/10.1002/2014JF003333
  - Thanks for pointing out this study. We will add the citation in a revised version of the manuscript
  - Added citation to L76
  - Added Dow et al., 2015 to references
- 3. L113: What is the ice thickness here?
  - ~980 m below North Lake (Das et al., 2008). We will add to L113 in a revised version of the manuscript.
  - Added to L131
- 4. L118: What is the baseline distance from KAGA?
  - ~55 km. We will add "KAGA base station on bedrock ~55 km away... (Bevis et al., 2012; Stevens et al., 2015)." to L118 in a revised version of the manuscript.
  - Added to L137
  - Added Bevis et al., 2012 to references
- 5. L122: Only 14 stations are shown in Fig 1, where is the 15th?
  - Station FL03 was inadvertently left off Fig 1; it is located on the western side of the array. We added its position to a revised version of Fig 1 in the pdf supplement of this comment response.
  - Added FL03 to revised Figure 1
- 6. L113-122: What is the uncertainty in the GPS station positions? (Horizontal and vertical)
  - Horizontal (vertical) 1-sigma errors are consistently +/-2 cm (+/-5 cm) across all stations and years (Stevens et al. 2015). We will add in a revised version of the manuscript.
  - Added to L138-139
- 7. L180: I suggest including the drainage basin outline in the study area figure.
  - Good suggestion. We added to a revised version of Figure 1 (pdf supplement).
  - Added to revised Figure 1

- 8. L245: Do you mean 168.85 is the end date for the pre-speed-up event? It looks like that date corresponds to the beginning of the orange bar in Figure 5. What is the start date for the pre-speed-up event velocity determination? It does not appear to be at the beginning of the x-axis shown in Figure 5 due to the location of the blue bars (calculated values) with respect to recorded velocities in that window (particularly at stations NL01, NL02, and NL06.
  - Correct, DOY 168.85 is the end date for the pre-speed-up event. The start date varies from 165 to 166 across the stations based on data availability at each sensor location. We will clarify this in a revised manuscript.
  - Revised L306 and revised Figure 5 x-axis
- 9. L290/Figure 7: Symbology on y-axis does not match text where the subplots are referred to as normalized
  - We will change reference in L290 from "(Fig. 7a and b)" to "(Fig. 7b)"
  - Revised L369
- 10. L318: Andrews et al., 2014 did not use in situ observations to show that channelization could account for decreasing velocities in the early melt season because they were only able to instrument moulins (to monitor the channelized system) during the middle of the melt season (between doy190-200 of each year) in mid-July.
  - Thank you for clarifying. We will revise L318 to "hypothesize" from "show."
  - Revised L398
- 11. L323: weakly-connected cavities are not "low-water pressure" because the channelized drainage system operates at lower pressures than even drained or hydraulically connected cavities. I suggest rephrasing to something like "these dewatered cavities maintain lower pressures than isolated cavities" or similar.
  - We will revise L323 to "these dewatered cavities maintain lower pressures than isolated cavities."
  - Revised L403
- 12. Fig 1: (a): The font size for the scale bar is too small to read. (b,d): What is the rationale for cutting off the x-axis bound on the date indicated? The last speed-up/melt event is close to that date so does it represent the end of the melt season or is there more variability after this point? (B,C) would benefit from the addition of the velocity at each individual station (in thin lines) that is then overlaid by the blue average line shown. This may be too visually cluttered and if so, I would appreciate a larger version in the supplement with this information as I am curious if some stations have a consisteently lower amplitude or higher amplitude response to the melt events.
  - See revised Figure 1 (pdf supplement) with increased scale bar font. We cut off the x-axis when winter velocities were established and melt had ceased and will note this in a revised Figure 1 caption. We also made a new figure (pdf supplement) showing all station velocities plotted on top of the average velocity across the array. Small systematic variations above and below the average array velocity can be seen in this figure (which will be added as a supplement to a revised manuscript). Note that Figure 11(pdf supplement) also displays the variability at each station and shows that in general stations in the north have lower amplitude and stations in the south have consistently higher amplitude response to melt events.
  - Revised Figure 1 and added Figure S1
- 13. L183: Could other moulins not be identified from satellite imagery (or field observations) to remove this area from the drainage basin?
  - It is challenging to identify all moulins across the region. Further, the input of melt from nearby moulins is likely contributing to the melt-induced speed-ups recorded across the North Lake GPS array. Thus, we feel the most robust estimate for the potential melt contributing to the velocity response, is to consider the average runoff across all grid cells in the drainage basin.

- 14. L216: melt events should be labeled in Figure 1 (or colored to correspond with in-text citation), this would make it easier for the reader to reference the figures from within the text.
  - Good suggestion. Please see the revised Figure 1 (pdf supplement), which shows the DOY 180 event in blue, representing a distal lake drainage (flood event). All other grey bars correspond with melt events.
  - Revised Figure 1
- 15. Figure 2: I suggest labeling the GPS stations to more easily compare with other figures (e.g., Figs 4,5). Also, this figure includes more stations than are included in the study area Figure 1.
  - We will add station labels in revised version of Figure 2. As noted above, the revised Figure 1 now shows the missing sensor (FL03).
  - Revised Figure 2
- 16. Figure 3: It would be helpful to emphasize the longer duration of the x-axis in subplots b and d for event 2011/238 by making the axis length scale the same as in plot a. I understand the purpose of the plot is to compare the displacement magnitude, however, it could be misleading to readers. At minimum I would suggest adding in daily minor tick marks to the x-axis in b&d.
  - We will add daily ticks to x-axis in subpanels b&d. Event duration of late season melt event is longer than the rapid lake drainage, so decreasing the duration shown on the x-axis would not capture the peak displacements.
  - Revised Figure 3 by adding daily ticks to panels b and d
- 17. Figure 4: The text within the figure is very difficult to read, consider increasing the font size and using a black rather than grey. (Same comment for Fig 5). I am not sure what stations or locations subplot label FL03 are referring to, a station with this name is not included in Figure 1 (maybe the 15th missing station?)
  - We will make font larger in revised version of Fig 4, 5, and 9 by reorienting the figures to consist of 3 columns and 4-5 rows. See the revised version of Figure 1 (pdf supplement) for the addition of sensor FL03.
  - Revised Figures 4, 5, 10 (Figure 10 was formerly Figure 9)
- 18. Figure 5: What is the length of time used to determine Vpre in this figure? From the figure it appears that it is <1 day of data, however, I think it is longer considering the methodology. I suggest extending the x-axis to show the full velocity window used to determine the Vpre value. I am not sure what stations or locations subplot label FL03 are referring to.
  - The time window varies station-to-station, most notably in the early season, when the sensors began collecting data over a range of days (between DOY 162 to 167) depending on their deployment. We will revise x-axis to a consistent start date of DOY 162; however, this will be prior to the collection of data for some stations. FL03 was added to a revised Figure 1.
  - Revised Figure 5
- 19. Figure 9: The axis for NL09 is different from the others, consider making them all uniform for an easy visual comparison of station response.
  - We will revise the y-axis to match 0–450 m/year in revised version.
  - Revised Figure 10 (formerly Figure 9)
- 20. Figure 11: Great figure, I am glad this analysis and visualization was included in the manuscript.
  - Thank you!

**Reviewer 2**

- 1. Firstly, I think the analysis of the correlation between runoff and speed-up magnitude in Section 3.3 could be improved. It is unsurprising that there is no correlation between runoff magnitude and speed-up magnitude, as it is the preceding melt conditions which "define" how much runoff the subglacial drainage system can accommodate (e.g., Hoffman et al. 2011). Hence why we see large dynamic responses from "spring-events" or supraglacial lake drainages, as they provide a large input of meltwater relative to the preceding period. This has been well established (e.g., Bartholomew et al., 2010; Hoffman et al. 2011; Schoof 2010). A suggestion would be to shift the focus away from runoff magnitude (mean, max and total) and instead focussing on rate of change of runoff or runoff variations from preceding conditions to the melt event and/or estimated lake drainage meltwater volume. I believe this is especially important, as currently the lack of a relationship between runoff magnitude and speed-up magnitude is one of the main findings, and this is already well known/ established.
  - Excellent suggestion. We agree that the preceding melt conditions may play a role in determining how much runoff the subglacial drainage system can accommodate and should be considered when interpreting the velocity responses. To address this, we created a new figure, incorporating the rate of change of runoff ( $\Delta R$  and  $\Delta R_n$ ) alongside speed-up ( $\Delta V$ ), relative speed-up ( $\Delta V_n$ ), and std. of  $\Delta V$  and  $\Delta V_n$ . We used the same methodology to estimate  $\Delta R$  (max runoff pre-event runoff) &  $\Delta R_n$  (max runoff /pre-event runoff) as was used to determine  $\Delta V$  &  $\Delta V_n$ . We find that there is a positive correlation between  $\Delta V$  &  $\Delta V_n$  and  $\Delta R$  &  $\Delta R_n$ , similar to the relationship between  $\Delta V$  &  $\Delta V_n$  and day-of-year (DOY) (Figure 7). We also observe a negative correlation between the standard deviation of  $\Delta V$  &  $\Delta V_n$  and  $\Delta R$  &  $\Delta R_n$ . Both figures can be found in the pdf supplement of this comment response
  - The results shown in the Rate of Runoff Analysis 1 (pdf supplement) highlight the interplay between the velocity response, changes in runoff, and seasonal changes in the hydrologic system (e.g., subglacial drainage state and the number of open moulins). However, a complication in separating the main factors driving the velocity response is the positive correlation between ΔR & ΔRn and DOY (Rate of Runoff Analysis 2, pdf supplement).
  - Because these variables are correlated with one another it is not surprising that they have similar relationships with  $\Delta V$  &  $\Delta V_n$ . However, masked in this relationship is the fact that  $\Delta R$  &  $\Delta R_n$  significantly underestimate the true runoff for the 3 rapid lake drainages (because the RACMO runoff estimates do not account for the meltwater stored in the lake basin). This implies that early season lake drainages have very large values of  $\Delta R$  &  $\Delta R_n$ , but small values of  $\Delta V$  &  $\Delta V_n$ , inconsistent with the correlations seen in the Rate of Runoff Analysis 1 figure. Thus, while we agree that changes in the rate of runoff may play a role in controlling the system response (particularly for runoff-driven events), we feel the temporal evolution of the melt system remains a key variable in the overall response of the ice sheet to meltwater forcing. We will address these issues more fully in a revised manuscript and propose including the figures above to highlight these points.
  - Added change in runoff figure, analysis, and discussion L450-494 (Figure 8, Figure S4).
- 2. In its current form, the novelty of the study is limited, as numerous previous studies have used GPS-derived ice velocities to infer changes in the subglacial drainage system as the melt season progresses. The data and methods have also been previously reported on, and I do not think this study provides new inferences about the subglacial drainage system evolution by applying the data, in the way that it does, to look at differences between melt-induced and lake drainage speedup events. The main aspect of this the study which could provide new insights on the evolution of the subglacial drainage system compared to previous studies, is the use of a dense array of GPS

stakes. This setup enables the examination of spatial variability in the dynamic response to melt events, which has been looked at in this study. However, the analysis of this aspect is currently limited and could be expanded upon to make it the main purpose.

- We concur that the main focus of our study is how velocity varies across individual events throughout the melt season. However, our study also analyzes the spatial variabilities across the GPS array during individual events and throughout the melt season, which has not been previously reported. We document that local lake drainages tend to have greater variability than regional melt events (Figures 4,5, and 9). Additionally, the sensors in the southern half of the array tend to have greater velocities compared to the north throughout the year, corresponding to a bedrock basin (Figure 11, pdf supplement). We will further emphasize the spatial variability across the array in a revised version of the manuscript.
- Furthermore, the novelty of this study comes from the characterization of a late season melt event at DOY 238. Although late season melt events have been previously observed, this study provides the first documented coherence of speed-up over the spatial scale of several lake basins. We also provide further constraint to the timing of hypothesized channel closure.
- Added additional spatial variability discussion on L19-20 and L646-748
- 3. I don't quite follow how you have compared the magnitude of runoff from the late-season melt events to the magnitude of meltwater delivered to the bed from the lake drainage. The meltwater flowing through the system during the late-season melt event will be the runoff from the calculated surface catchment, plus all the runoff entering the subglacial system upstream of this (especially as you state the events last multiple days). The runoff is likely higher than you estimate, but still less instantaneous than the lake drainage. (L203)
  - The magnitude of meltwater delivered to the bed during the lake drainage was determined from the known volume of the North Lake basin and its inferred area of basal distribution. By dividing the volume of the lake basin by the area of the subglacial "blister," we were able to deduce a "effective thickness" (mm of w.e.) of melt at the bed. Further, to directly compare this value to daily RACMO data, we converted the 5-hour duration of the lake drainage to a 24-hr time period. In the middle-to-late season, we utilize RACMO runoff (mm of w.e. per day) to estimate the melt throughout the season. We calculate the average runoff across the drainage basin that North Lake resides in.
  - Additionally, as discussed above, we determined the rate of change in runoff using the same methodology to estimate velocity and relative response:  $\Delta R$  (max runoff preevent runoff) &  $\Delta R_n$  (max runoff /pre-event runoff). The pre-event runoff is an average daily runoff across the pre-event time period described in the manuscript.
  - New change in the rate of runoff methodology defined on L452-457
- 4. The structure and coherence of the writing in the manuscript could be tighter. Currently, there is a lot of repetition of the main findings and introduction paragraphs throughout (e.g., L92-101, L309-306). To improve the readability and impact, the ideas presented in the discussion could be also be condensed.
  - Thank you for this suggestion to improve the clarity of the manuscript. We will condense text throughout Introduction, Results, and Discussion, notably in L92-101 and L306-309 in a revised version of manuscript.
  - We have streamlined manuscript and reduced repetition throughout the revised manuscript
- 5. I am slightly confused how Section 4.2 in the discussion fits into the rest of the manuscript, as it seems to come out of nowhere.
  - We apologize for the abrupt transition. The latter half of Section 4.1 utilizes annual trends across the GPS array (Figure 7) to characterize and infer the timing of the distinct

phases subglacial drainage system. However, the DOY 180 event (described in Section 4.2) consistently has the largest variability, which diverges from the annual trends described in Section 4.1. Thus, the goal of this section was to further explore the origin of this variability, and in doing so we for the first time identify the response of a local GPS network to a distal lake drainage event. We will better motivate this discussion in a revised manuscript.

- The transition to Section 4.2 is on L381-384. After describing the general trends of Figure 7, we point out the outlier on 2012/180, which we state will be discussed further in Section 4.2. This is an appropriate transition as it is prior to the discussion, and after the anomaly event is identified in the results.
- 6. Title: For consistency with the manuscript and other literature, perhaps "subglacial hydrologic system" can be changed to "subglacial drainage system".
  - We will update in a revised version of the manuscript.
  - Revised to subglacial drainage system throughout manuscript.
- 7. L19: Change "transients" to "transient speed-up events" or similar
  - We will update in a revised version of the manuscript.
  - Revised velocity transients to transient speed-ups throughout manuscript.
- 8. L23: I'm not sure I follow what you mean by "basal transmissivity" here, can you instead say "increasing basal friction" or similar. Also, the reason for the large magnitude speed-up during this time of year, is the reduced capacity of the subglacial drainage system to "handle" this sudden extra melt, not the fact that an increase in basal friction has led to lower ice velocities. Perhaps these sentences could be rephrased to make this clearer.
  - "Basal transmissivity" (or "basal hydrologic transmissivity") refers to "the ability of the meltwater to move through the basal hydrologic system" (L57) and is formally defined as the hydrologic conductivity multiplied by the saturated layer thickness (Lai et al., 2021). Thus, "basal transmissivity" does not refer to basal friction and a decrease in basal transmissivity is consistent with the understanding that in the late-season the subglacial system has a reduced capacity to handle a sudden influx of extra melt. We will add the formal definition to a revised version of the manuscript to clarify this concept.
  - Added formal definition to L62-64
- 9. L45: Change "ice-sheet" to "basal"?
  - We will update in a revised version of the manuscript.
  - Revised L46
- 10. L46: Here and multiple other times you use "basal/subglacial" "drainage/hydrologic" interchangeably. Can you please stick to just one? I suggest just using "subglacial drainage system".
  - We will update to "subglacial drainage system" in a revised version of the manuscript.
  - Revised to subglacial drainage system throughout
- 11. L56-60: This sentence is quite long and hard to read, can it be rephrased?
  - Agreed. We will revise to "These observations suggest that the hydraulic transmissivity (i.e., the ability of the meltwater to move through the basal hydrologic system) becomes more efficient beneath the lake as the melt season progresses (Lai et al., 2021). Furthermore, these findings are consistent with model predictions (Schoof, 2010) and observations (Chandler et al., 2013; Andrews et al. 2014; Andrews et al. 2018) premised on a seasonal evolution towards a more channelized subglacial meltwater system with increasing meltwater input (e.g., Schoof, 2010)."
  - Revised L62-66
- 12. L70: Could you provide a timescale for creep closure of subglacial channel closure (e.g., from Chandler et al., 2013)? It will help with the interpretation in your discussions

- We will add "on timescales of hours to days (Chandler et al., 2013)" in a revised version of the manuscript.
- Revised on L76
- 13. L78: With no evidence, you can't be sure they are an "order-of-magnitude smaller" so perhaps remove. The main differences are the rate of meltwater delivery between the two.
  - The reviewer is correct that the rate of meltwater delivery between the two is the main difference. However, this line is referring to the volume of melt difference between the melt stored in the North Lake basin and the magnitude of melt reaching the bed from a regional melt event. We will revise to "smaller" as this discrepancy is not always an order-of-magnitude, depending on if you consider the maximum, mean, or integrated melt "thickness" (proxy for volume). See below for these reference values; also shown on Figure 6 of the manuscript.
  - Lake Drainage Effective Runoff (mm of w.e./day): 1650-3000 (Max), 110-200 (Mean), 330-600 (Integrated)
  - Regional Melt Events Runoff (mm of w.e./day): 13.9-60.1 (Max), 7.6 to 40.3 (Mean), 70.8 to 273.8 (Integrated)
  - \*We correct for the difference in the rate of delivery by converting all runoff to daily timescales.
  - Revised to "smaller" from "order-of-magnitude smaller" on L100
- 14. L89: Can you state the distance from the terminus?
  - $\sim$ 25 km. We will add in a revised version of the manuscript.
  - Added distance on L113
- 15. L91: I know that the NIF is described in the methods, but because it is not a commonly used method, perhaps brief detail on what it does could be added here, e.g., "we use a NIF to infer basal slip..."
  - Good point. We will add details on our use of the NIF in this paper, such as, "we use a Network Inversion Filter (NIF) (Stevens et al., 2015) to infer the basal slip and basal uplift of a late-season, transient speed-up event and compare to an early-season lake drainages at the same location."
  - Revised L114-116
- 16. L97: Rephrase "ice-response indicators"
  - We will update in a revised version of the manuscript.
  - Revised to "values" on L118
- 17. L101: This sentence is largely a repeat of what has previously been said in this paragraph. I wonder if this paragraph can be cut down, and instead changed to focus on what your objectives are. L92-101 currently reads more like a conclusion.
  - We will remove L93-95 and L97-100 to shift focus from findings to objectives. L101 refers to our hypothetical model construction, which has not yet been mentioned in the paragraph.
  - Removed findings from paragraph L118-126 so that it only describes the objectives
- 18. L101: The literature review provided in the introduction is clear, but I would suggest also referencing Schmidt et al. 2023 and Hoffman et al. 2011, which are currently not in the reference list but are two very relevant papers to this study.
  - Thank you for this suggestion. We will incorporate references in a revised version of the manuscript.
  - Added Hoffman et al 2011 to references and L56, L58, L64-65, L390, L446
  - Added Schmid et al 2023 to references and L44, L65, L103-104
- 19. L175: What is the timestep of the RACMO data, hourly or daily?
  - Daily, we will state the timestep in the revised manuscript.

- Added "daily" to L223
- 20. L181: Change "points" to "grid cells"
  - We will update in a revised version of the manuscript.
  - Revised L226-227
- 21. L181: Surely you need to calculate the sum of all grid cells in the catchment to get a measure of the total runoff entering the subglacial system?
  - Yes, we sum all grid cells in the catchment then divide by the number of cells. This average across the catchment basin is significantly smaller than the runoff collected in the North Lake basin that rapidly drains during a lake drainage event. As part of the revisions described above, we use this approximation to determine the pre-event runoff by averaging the daily runoff across the time-period preceding the speed-up event. We found the maximum event runoff during the speed-up event. The maximum runoff and pre-event runoff were used to determine the rate of runoff (as described above).
- 22. L190: Change "total" to "integrated" to keep consistent with your plots, or vice-versa
  - We will update to "integrated" in a revised version of the manuscript.
  - Revised L235
- 23. L218: The phrase "plotted relative to the onset of the speed-up event" is confusing as it sounds like you have plotted basal slip relative to the timing of the speed-up onset. I believe you have actually plotted it relative to the basal slip before the event? Although in the caption for figure 2 you state that it is the maximum basal slip of each event? Please can you improve the clarity of this sentence and the caption of Figure 2.
  - Sorry for this confusion. A pre-speed-up event time period is needed to define the background ice speed and direction for the NIF. Figure 2 then shows the maximum extra basal slip relative to this background ice velocity (Stevens et al., 2015). In other words, if the ice were to continue to move at exactly the same rate as during the pre-speed-up event period, the NIF would report no "extra basal slip". This will be clarified in a revised manuscript.
  - Revised L268-269
- 24. L261-268: This paragraph is a repeat of the previous two and could be omitted
  - Good suggestion. This paragraph will be removed as it is expanded upon in the discussion as well.
  - Removed paragraph on L321-328
- 25. L276: Can you just say "average delta V" here and throughout to make it easier for the reader to follow? And perhaps rephrase "does not exceed" as you are referring to an average not individual data points.
  - $\Delta V$  is a standard notation for average that is also used in Figure 11 when we define  $\Delta V_{\text{array}}$ .
- 26. L311: Delete "Recently,"
  - We will update in a revised version of the manuscript.
  - Removed word on L391
- 27. L336: I am not sure you can be confident the volume of meltwater is less, just that the delivery is less instantaneous than during a hydro-fracture lake drainage.
  - See L78 response (#13). The delivery is less instantaneous; however, we also provide an approximation of the melt volume delivered to the bed at a lake drainage and a melt event. We adjust for rate of delivery by converting to daily timescales, including the rapid lake drainage. Specifically, the DOY 238 event has notably less melt than the lake drainage as shown by the "thickness" of melt at a lake drainage and as averaged across the drainage (i.e. at any given grid cell across the catchment basin):

- Lake Drainage Effective Runoff (mm of w.e. per day): 1650-3000 (Max), 110-200 (Mean), 330-600 (Integrated)
- DOY 238 Runoff (mm of w.e. per day): 29.5 (Max), 8.8 (Mean), 70.8 (Integrated)
- 28. L337: Again, I think it is the variation in meltwater delivery to the bed that is important here, not the total runoff/drainage volume
  - See discussion above regarding our new analysis of the sensitivity of the ice sheet velocity response to changes in the rate of meltwater delivery.
- 29. L344: It would be good here to give a measure of the amount of runoff in the preceding period before the speed-up event, as this is the most important factor in inducing the dynamic response. Additionally, do you have an estimate for the ice thickness under the GPS array it would be helpful to add this information to your introduction.
- 30. Important to highlight that the runoff drops close to zero for a prolonged period before the late-season melt event. Basically, a spring event but with an established supraglacial drainage system.
  - We will address runoff preceding speed-up with the revised rate of runoff figures (pdf supplement) and will add ice thickness (~980 m below North Lake (Das et al., 2008) to introduction in a revised version of the manuscript.
  - ~980 m ice thickness added to L131
  - Runoff methodology, discussion, and figure added (per RC2 revision #1)
- 31. L347: I would argue that the subglacial drainage efficiency is a result of the preceding runoff magnitude. Please provide a reference(s) for the hypothesis you are referring to.
  - We will add Schoof (2010), Hoffman (2011), and Hewitt (2013) references, which emphasize the role of the subglacial system in describing sliding behavior.
  - Added references to L445-446
- 32. L354: Can you clarify what you mean by the "style" of runoff delivery? Is this supraglacial lake drainage vs normal runoff routing through moulins?
  - Sorry for the confusion, but your understanding is correct. Here "style" refers to a rapid supraglacial lake drainage injecting a large volume of melt to the basal system in a few hours versus the mid-late melt season events, which consist of normal runoff routing through moulins on longer timescales. We will clarify this in a revised manuscript.
  - Revised L497
- 33. L340: Change "basal" to "subglacial"
  - We will update in a revised version of the manuscript.
  - Replaced on L436
- 34. L385: This study does not report on annual ice motion at North Lake
  - We will revise "as noted" to "similarly." We calculate the displacement contribution of
    the late season melt event to annual ice motion at North Lake to directly compare our
    results at North Lake to the Ing et al. (2024) findings regarding the relatively small
    impact of late-season speed-ups on annual ice discharge in western Greenland.
  - Revised L434
- 35. L391-407: The conceptual model presented here is largely a repeat of models that have already been established (e.g., Hoffman et al. 2016; Davison et al. 2019), so I question whether this paragraph and Fig. 8 provides anything new and is necessary?
  - Our model emphasizes the role of many open moulins in the late season, and couples this with the closure of channels. Thus, we emphasize the role of a small melt volume allowing for significantly larger, yet uniform, velocity response in the late season.
- 36. L392: Change Figure 8 to Fig. 8?
  - We will update in a revised version of the manuscript.
  - Revised on L542 (now Fig. 9)
- 37. L402: Remove "ice sheet" as the processes you are describing here are specific to your site (lake drainages)

- We will update in a revised version of the manuscript.
- Removed on L552
- 38. L409-413: I'm not sure if there is anything novel added here.
  - We are describing our results in the context of the ideas of Andrews et al. (2014) and Hoffman et al. (2016) and defending our reasoning for the closure of channels around DOY 238. Evidence of the closure of channels in this region has not been previously described.
- 39. L460: I'm not sure these results provide "preliminary insights", with many previous studies reporting on the evolution of the subglacial drainage system in this area.
  - Previous studies in this area do not describe the subglacial drainage system beyond DOY 210. We provide preliminary insights of the drainage system beyond that observable via a lake drainage study.
- 40. L461-463: This sentence is hard to read at present, can it be rephrased?
  - We will revise to "We find enhanced ice-flow sensitivity to melt input in the form of longer, more uniform, velocity responses during late-season melt events compared to early- to mid-season lake drainage or melt events."
  - Revised on L633-635
- 41. L471: Isn't this just because there are more of them?
  - No, this was determined considering individual events. Each individual melt event, generally, has a longer duration and greater velocity response amplitude (seen in Fig. 7a and b).
- 42. L680 (Figure 2): It is difficult to distinguish the black GPS triangles from the black arrows, perhaps you could use a different colour?
  - We will change GPS triangles to grey.
  - Revised Figure 2
- 43. Figure 1: Please add lat/lon coordinates to the border of (a) and other figures where appropriate. Change y-axis of (d and e) to Daily runoff (mm w.e.). Change units in (b) and (c) to m year-1. This and for all figures change "a)" to "(a)".
  - Please see the revised version of Figure 1 (pdf supplement). We added lat/lon coordinates to the border of panel (a), changed the y-axis (d,e) to "Daily runoff (mm w.e.)," and "m year-1," and revised labels to "(a)." We will implement similar changes to Figures 10 and 11.
  - Revised Figure 1
- 44. Figure 6: I find Figure 6 somewhat hard to interpret. I believe the point of this figure is to relate change in velocity of speed-up events to the mean runoff of the event, the maximum runoff of the event and the total runoff of the event. All of which show little or no correlation. I wonder whether comparing the change in velocity to all of the mean, total and max runoff is necessary. What seems lacking is the comparison to the change in runoff, as this is the main factor that causes the speed-up (see Hoffman et al. 2011)
  - Your understanding is correct, we find little or no correlation between change in velocity and various measurements of event runoff. We address the change in the rate of runoff in a revised version of Figure 6 (pdf supplement).
  - Added Figure 8
- 45. Figure 8: Can you make the figure labels (a, b, etc) bigger, they are currently hard to read.
  - We will make figure labels larger in a revised version of Figure 8.
  - Revised Figure 9 (formerly Figure 8)
- 46. Figure 10: Here and throughout change "m/year" to m year-1 or m yr-1. Please add coordinates to map border.
  - We will revise to m year-1 and add coordinates to map borders in a revised version of Figure 10 (similar to revised Figure 1 and Figure 11).

- Revised Figure 1, 11, 12 (formerly Figures 10 & 11)
- 47. Figure 11: Please increase font size of figure labels, scale bar and legend in b. Please can you move the figure labels to the left of each plot.
  - We increased font size of figure labels, scale bar, and legend (b), and moved the (a) and (b) labels to the left side of the plots in a revised version of Figure 11 (pdf supplement).
  - Revised Figure 12 (formerly Figure 11)

---

## Referee Report (RR1)

**General comments:**

I thank the authors for implementing my suggested changes and answering my queries. I still have a few comments and questions surrounding the runoff methodology and the new rate of runoff analysis.

Additionally, I still believe the writing could be improved further. I have made numerous suggestions below on where sentences could be rephrased to be clearer. The language used should also be more consistent. The authors interchangeably use varied vocabulary to describe the same thing (e.g., speed-ups vs ice acceleration, velocity transients; subglacial drainage system vs meltwater system, basal system, basal channels). I recommend going through the manuscript to make the writing clearer and to be more consistent as it will greatly help to improve the readability of the manuscript.

**Runoff methodology:**

The authors state that they calculate runoff for late-season melt events by summing all grid cells in the catchment and then dividing by the number of cells to get the average runoff (not the total runoff for the catchment). However, as mentioned in my initial review, I'd like to clarify that the correct approach is to simply sum the grid cells of the whole catchment which will give the total runoff from that catchment entering the subglacial system below that catchment (assuming all water directly accesses the bed and there is no upstream subglacial influence – see later comments).

Multiple times in the manuscript the authors compare the magnitude of the late-season melt events to that of the early season lake drainage, whilst stating themselves that this is a generalised estimate that ignores any upstream influence (likely substantial). Given that the catchment-based runoff calculation is an underestimate (especially when using the average and not the total), I question whether a direct comparison between the two event types is justified. To improve the robustness of the runoff estimates, the authors could route RACMO using the hydropotential to the subglacial area affected by the North Lake drainages. Due to the large uncertainties in both RACMO and bed topography and the extra work required this isn't essential, but without a more thorough method to calculate the runoff, I recommend removing all comparisons in runoff magnitude between the lake drainage and the late season melt events.

**Rate of runoff analysis:**

Thank you for adding the rate of runoff analysis to the manuscript. I do wonder if this analysis would be more suited to being in the results with the rest of your runoff analysis?

L484-488: I'm a bit confused what point the authors are trying to make here. It's interesting that there is a positive correlation between  $\Delta R$  and DOY, which is exactly why late season melt-induced speed-ups are of interest and why they trigger large speed-ups compared to melt events during peak melt season (i.e., because the rapid increase in runoff compared to preceding periods overwhelms the subglacial drainage system). Why does this add a complication?

Additionally, I don't quite follow the authors comment about the rate of change of runoff in Section 4.1: "Thus, while we agree that changes in the rate of runoff may play a role in controlling the system response (particularly for runoff-driven events), we feel the temporal evolution of the melt system remains a key variable in the overall response of the ice sheet to meltwater forcing.", The rate of runoff (melt, rainfall, lake drainages) is what predominately controls the temporal evolution of the

drainage system (e.g., Schoof 2010; Hoffman et al. 2011, Bartholomew et al. 2011, etc). If you have high sustained melt going into the system, it will get efficient and respond less to melt or lake drainage events. This is why the lake drainages early in the melt season have large dynamic response. I recommend carefully rewriting this section to explain the points made in more detail, whilst also referring to the well-established concepts of subglacial drainage evolution in Greenland.

**Section 4.2:**

I believe this section still doesn't fit in with the rest of the main manuscript, and complicates the overall story. The influence of bed topography channelling an upstream lake drainage is very specific to this site. It does not fit in with the general inferences of the evolution of the subglacial drainage system through the use of trainset speed-up events presented in the rest of the manuscript. Moreover, it also highlights the flaw in the runoff estimates, that the subglacial drainage system beneath North Lake is well connected to upstream sources. I recommend removing this section to streamline the manuscript and to help present a clearer story.

**Specific comments:**

Title: Seasonal subglacial drainage system evolution? The authors talk about surface drainage too, but I wonder as the focus is on the inferred evolution of the subglacial drainage system it is worth mentioning this in the title.

L35: ...(GPS) observations of ice motion show that...

L36: Correct van de Wal et al. 2008 reference. I also recommend diversifying references for this bit (e.g., Andrews et al., 2014; Bartholomew et al., 2011)

L38: Delete "the details of". Change "ice-sheet velocity" to "ice velocity"

L39: Add "is often non-linear"

L44: Suggest changing the end of this sentence to something similar to: "...varies throughout the melt season as subglacial drainage transitions from inefficient to efficient systems, modulating basal sliding. + references"

L46: Change "basal" to "ice"

L48: Change "...how the ice sheet responds to..." to "how ice velocities respond to...". Change "ice-sheet sliding" to "basal sliding".

L51: Please add example references for supraglacial lake drainage studies

L52: Supraglacial lakes aren't limited to the western margin, change to/or similar "In the ablation zone of the Greenland Ice Sheet..."

L56: Remove "glacial". Change to "reduces friction between the ice and bedrock.."

L57: Delete "the" from "the lake drainage events...". Delete "these". Change to "...coincide with surface uplift driven by high water pressures in the subglacial drainage system"

L59: I might be being pedantic here, but to me "ice sheet" refers to ice sheet wide processes, whereas lake drainages are local/regional. Perhaps say "ice"?

L62: The added definition is much appreciated here, perhaps change to saturated layer thickness at the ice-bed interface?

L68: Change to "surface uplift"

L71: Change to "subglacial drainage efficiency"

L75: Basal channels more commonly refer to channels under ice shelves. Suggest rephrasing sentence to/or similar: "Ice thickness also plays a role, with subglacial channels under thick ice (define thickness) closing quickly (within hours to days) through ice creep..."

L78: Delete "These observations highlight the need for further study on the evolution of basal conditions." Or change "basal conditions"

L81: "ice-sheet speed up" suggests they are occurring over the whole ice sheet. Perhaps delete "ice-sheet". And same for L82?

L100: Suggest adding "localised lake drainage". I'm still not entirely convinced you can confidently state that melt and rainfall events are smaller than lake drainage. With melt/rainfall events happening on much larger spatial scales, the increase in subglacial discharge for well-defined outlets will surely be larger than lake drainages. I suggest instead emphasizing the different spatial scales (local vs regional).

L101: Suggest changing to "transient ice velocity response to meltwater inputs...for annal ice motion". I would suggest refraining/being more careful about the use of "ice-sheet velocities" throughout, with the studies and processes you discuss in this study are all regional/local scales.

L103: Again, remove "ice-sheet"

L108-109: Suggest removing this line, as these two types of events operate on vastly different spatial scales.

L106-120: Good justification for the study, and improved intro to the study site.

L113: Is the site 25 km away from the terminus of Jakobshavn Isbrae, or another glacier?

L113: Change "ice-sheet" to "ice motion"

L139: Suggest rewording 1-sigma errors to 1-standard deviation?

L223: Are you using the daily mean runoff or daily sum runoff from RACMO?

L224: Add "ice surface catchment in which..."

L226: Shouldn't this be the sum of all grid cells? How many grid cells is the catchment?

L228: Change to/or similar "This is a generalized estimate for the runoff that makes it to the bed directly below the lake, but neglects any upstream sources routed beneath North Lake through the subglacial drainage system"

L235: Change "integrated" to "total"

L254: Again, I don't think these claims can be made without more thorough methods.

L237: Can you please add a reference for the timescale of North Lake drainage?

L307: pre-melt season winter velocity? Perhaps for consistency just say "background winter velocity"

L332: Change "Discussion section" to "Section 4.?"

L332: The analysis on runoff variability will be more suited here, in the results.

L336: Suggest changing "ice sheet response" to "ice velocity response"

L329: It will likely not make much difference, but I do think the effective lake drainage melt supply to the system should be your calculated effective runoff + RACMO runoff of the catchment for that day?

L389: Suggest rephrasing "the sliding behaviour of the Greenland Ice Sheet..." to/or similar "The relationship between ice velocities and surface melt are linked through the evolution of the subglacial drainage system."

L396: Change "basal hydrologic system" to "subglacial drainage system"

L427: I don't think you can confidently state it is smaller.

L434-435: I find this sentence confusing, please can it be rephrased?

L436: Change "subglacial meltwater system" to "subglacial drainage system"

L448: What is the maximum, mean or integrated flux? Do you mean runoff?

L450: Change "speed-up response" to "magnitude of speed-up" or similar

L451: Delete "summer"

L452-457: This should be in the methods instead

L445: This sentence adds confusion, with the "state" of the subglacial drainage system during the melt season being almost completely controlled by the runoff magnitude and variability... (e.g., Schoof, 2010; Hoffman et al., 2011 etc)

L450-482: I think this paragraph would be better placed in the results with the other runoff analysis, or possibly replacing the analysis of speed-ups compared to mean, max, etc runoff.

L479: Be careful with the use of "melt events", as I presume you mean peaks in runoff, which could also be caused by rainfall events?

L484: I suggest rewording to "The correlations between speed-up magnitude, DOY and the rate of change of runoff..."

L485: Change "subglacial drainage state" to subglacial drainage efficiency

L489: I would suggest rewording/removing. RACMO does not underestimate runoff from lake drainages – it doesn't include them at all.

L496-497: Yes, but it is the preceding runoff characteristics that define the "state" of the subglacial drainage system. And if there were a large runoff event before the early-melt season lake drainage, the velocity response would also change.

L508: Delete "glacial"

L509: Change "ice sheet" to ice

L536: Change "ice-sheet" to "ice"

L542: Confusing sentence, smaller amplitudes in what? Suggest rephrasing "horizontal sliding transients", surely there are no velocity speed-up events before the lake drainage?

L543: remove "the" ...water filled cavities. And network instead of networks?

L45: Change "velocity transient" to "speed-up"

L547: Change to: early in the melt season.... Change "hydrologic system" to "subglacial drainage system"

L547: Change "the horizontal-velocity increases..." to "Increases in ice velocity associated with..."

L549: Suggest changing "meltwater conduits" to "connections"

L549: More drainages occur? Or is there simply just more active supraglacial hydrology (moulins etc).

L551: I think this sentence could do with being more nuanced and specifying that it increases frictional coupling in the distributed regions adjacent to efficient channels?

L543: Timescale of closure is very dependent on ice thickness

L543: Typo? "dewater cavities"

L555: Change conduits to "connections" or just say moulins

L631: Change "subglacial bed conditions" to subglacial drainage system

L632: Suggest removing "preliminary" considering the wealth of previous papers on this site and many other studies using GPS to infer subglacial drainage evolution in west Greenland.

L645-648: Again, this doesn't fit with the rest of the conclusion/ story on the manuscript

L915 (Figure 6): Please check axis labels here and throughout (m year-1 should be m year-1). I think mm/day should also be mm w.e. day-1

Figure 1: Check axis labels throughout (m year-1 should be m year-1). How is the annual ice flow direction is indicated?

Figure 4: Correct m year-1

---

## Author Response (AR2)

Black Text – original comment from reviewer (line #s correspond to previous draft)

Red Text – edits made to revised version of manuscript (line #s correspond to revised version of manuscript with tracked changes)

**General comments:**

I thank the authors for implementing my suggested changes and answering my queries. I still have a few comments and questions surrounding the runoff methodology and the new rate of runoff analysis.

Additionally, I still believe the writing could be improved further. I have made numerous suggestions below on where sentences could be rephrased to be clearer. The language used should also be more consistent. The authors interchangeably use varied vocabulary to describe the same thing (e.g., speed-ups vs ice acceleration, velocity transients; subglacial drainage system vs meltwater system, basal system, basal channels). I recommend going through the manuscript to make the writing clearer and to be more consistent as it will greatly help to improve the readability of the manuscript.

We thank reviewer for their detailed comments and have tried to address their concerns in our revised manuscript. Below we describe our changes in response to each of their individual comments.

**Runoff methodology:**

1. The authors state that they calculate runoff for late-season melt events by summing all grid cells in the catchment and then dividing by the number of cells to get the average runoff (not the total runoff for the catchment). However, as mentioned in my initial review, I'd like to clarify that the correct approach is to simply sum the grid cells of the whole catchment which will give the total runoff from that catchment entering the subglacial system below that catchment (assuming all water directly accesses the bed and there is no upstream subglacial influence – see later comments).

The reason, we do not elect to sum the total runoff of the catchment basin is because the short duration of the speed-up events (typically < 5 days) makes it unlikely that all melt across the 5,324 km2 catchment basin is routed to North Lake on a timescale relevant to the observed speed-ups. This recommendation by the reviewer, however, did prompt us to change our methodology to be less generalized, and to use only the RACMO runoff points closest to North Lake rather the runoff average across the catchment basin. We revised our methodology to average the 6 closest RACMO grid cells to North Lake that are within the catchment basin. In doing so, we consider only the runoff within ~33 km of North Lake. This assumption is based on the velocity of subglacial flow estimated by prior studies in regions with similar conditions. For example, western Greenland lake drainage flood waters have been observed to flow at speeds ~0.3 m/sec (26 km/day) (Hoffman et., all 2011; Stevens et al., 2022). Further, subglacial water flow under ice of similar thickness (~1000 m) has been observed to travel at speeds of 0.2 m/sec (17 km/day) (Chandler et al., 2013). Thus, we estimate that during a single speed up event melt could flow distances of several 10s of km through the North Lake subglacial system, but is not likely to flow over distances required to pool runoff from the entire catchment. Further, we found that varying the number of grid cells used to estimate the average

runoff feeding the subglacial system did not significantly alter our runoff estimates or the observed correlations between speed up and runoff (Fig. S2).

We added a discussion of subglacial meltwater speeds to motivate this methodology on L225-231. Further, we added a supplemental figure (S2), which compares the 1-point, 6-point, and 44-point (original) average to a single point closes to North Lake. The singular point and 6-point averages are very similar and are consistently larger than the 44-point average (which includes many higher elevation points), thus motivating this methodology revision. We stress however that none of the major trends discussed in the original manuscript have changed.

2. Multiple times in the manuscript the authors compare the magnitude of the late-season melt events to that of the early season lake drainage, whilst stating themselves that this is a generalised estimate that ignores any upstream influence (likely substantial). Given that the catchment-based runoff calculation is an underestimate (especially when using the average and not the total), I question whether a direct comparison between the two event types is justified. To improve the robustness of the runoff estimates, the authors could route RACMO using the hydropotential to the subglacial area affected by the North Lake drainages. Due to the large uncertainties in both RACMO and bed topography and the extra work required this isn't essential, but without a more thorough method to calculate the runoff, I recommend removing all comparisons in runoff magnitude between the lake drainage and the late season melt events.

We elected to revise our methodology as described above, given the short duration and location of our speed-up event analysis. We reiterate that we do not think a summation is justified as this would significantly overestimate the melt entering the North Lake subglacial system. Rather, we feel the average runoff from the local area provides a good estimate of the rate that water (mm w.e. per day) is being input into the subglacial system during the speed-up event. In the case of the lake drainages, the "effective runoff" (again calculated in mm w.e. per day) is well known from the volume of the lake basin, the lake drainage time, and the areal extent of the basal melt distribution (known from previous modeling of the local uplift). This allows us to make a direct comparison between the lake drainage volume and the RACMO melt estimates as both are in units of mm (thickness) per day. Although there is some uncertainty with this value, it is justified to say that the runoff magnitude from a lake drainage is much larger than the runoff associated with the late season melt events, especially given the rapid nature of the lake drainage (hours compared to days for the runoff events). These estimates for the lake drainages are defined as "effective runoff" and are denoted in purple (Figure 6) to distinguish these values from the RACMO estimates for the same events. As Figure 6 contains three known lake drainage events, quantifying the discrepancy between the RACMO runoff estimate and the melt stored in the lake is especially important to contextualize our results.

**Rate of runoff analysis:**

3. Thank you for adding the rate of runoff analysis to the manuscript. I do wonder if this analysis would be more suited to being in the results with the rest of your runoff analysis?

We agree that the methods of the rate of runoff analysis better fit in the methods section and this description has been moved to Section 2.2 (L247-252). We kept Fig. 8 and its subsequent discussion in Section 4.1 as we feel this is where it best aligns with the flow of the paper. The rate of runoff analysis was prompted by our results shown in Figures 6 and 7, which further nuanced our discussion on the controls of velocity response throughout the melt season.

- 4. L484-488: I'm a bit confused what point the authors are trying to make here. It's interesting that there is a positive correlation between ΔR and DOY, which is exactly why late season melt-induced speed-ups are of interest and why they trigger large speed-ups compared to melt events during peak melt season (i.e., because the rapid increase in runoff compared to preceding periods overwhelms the subglacial drainage system). Why does this add a complication? Our word choice may have led to confusion. We do not think it is possible (given our methods) to completely deconvolve DOY from the various factors influencing speed-up (i.e., multiple variables such as the number of open moulins and the subglacial drainage system presence of channels and/or cavities and the extent of their "connectedness" have all been shown to change over the course of the melt season). Further, we know that
  - (i.e., multiple variables such as the number of open moulins and the subglacial drainage system presence of channels and/or cavities and the extent of their "connectedness" have all been shown to change over the course of the melt season). Further, we know that  $\Delta R$  does not accurately reflect the melt stored in the lake basin and supplied to the bed during a lake drainage. Indeed, if we were to take into account the lake volume, the relationship between  $\Delta R$  and DOY is substantially weaker. Thus, it seems in the early season the velocity response is predominantly predicted by the number of open moulins and the lack of efficient channels.
- 5. Additionally, I don't quite follow the authors comment about the rate of change of runoff in Section 4.1: "Thus, while we agree that changes in the rate of runoff may play a role in controlling the system response (particularly for runoff-driven events), we feel the temporal evolution of the melt system remains a key variable in the overall response of the ice sheet to meltwater forcing.", The rate of runoff (melt, rainfall, lake drainages) is what predominately controls the temporal evolution of the drainage system (e.g., Schoof 2010; Hoffman et al. 2011, Bartholomew et al. 2011, etc). If you have high sustained melt going into the system, it will get efficient and respond less to melt or lake drainage events. This is why the lake drainages early in the melt season have large dynamic response. I recommend carefully rewriting this section to explain the points made in more detail, whilst also referring to the well-established concepts of subglacial drainage evolution in Greenland.

In reality, changes in runoff cannot fully explain the velocity response throughout the melt season, likely due to the fact that the subglacial landscape is influenced by other non-runoff related factors, such as the distribution of open moulins, basal topography, and the initial runoff needed to establish channels in the early melt season. We discuss these factors in the results/discussion as we do not find that rate of runoff is predominantly controlling subglacial efficiency at all points in the melt season. This can be seen in our Figure 8 regressions. Further, Figure 8 does not take into account the "effective runoff" associated with the lake drainage events. If we consider the additional melt entering the North Lake subglacial system during a lake drainage, the effective values of  $\Delta R$  and  $\Delta R_n$ , would be much larger. For example, if we consider the effective

runoff of the DOY 169 lake drainage (~110 mm/day on average), and divide/subtract it by the pre-event mean rate of runoff (26.6 mm/day), it results in a larger  $\Delta R = 83.4$  and  $\Delta R_n = 4.1$  compared to  $\Delta R = 8.8$  and  $\Delta R_n = 1.3$  using the RACMO runoff estimate of 35.4 mm/day. This complicates the relationships shown in Figure 8 and suggests that the lake drainage events actually have rather small dynamic responses compared to their effective runoff. As shown in the schematic in Figure 9, we argue this is due to the lack of open moulins in the early melt season and more localized introduction of meltwater to the bed, leading the strong ice-bedrock coupling in adjacent regions. We have revised the text on L448-450 to clarify this point.

**Section 4.2:**

6. I believe this section still doesn't fit in with the rest of the main manuscript, and complicates the overall story. The influence of bed topography channeling an upstream lake drainage is very specific to this site. It does not fit in with the general inferences of the evolution of the subglacial drainage system through the use of trainset speed-up events presented in the rest of the manuscript. Moreover, it also highlights the flaw in the runoff estimates, that the subglacial drainage system beneath North Lake is well connected to upstream sources. I recommend removing this section to streamline the manuscript and to help present a clearer story.

We elect to keep Section 4.2 in the manuscript because this section attempts to explain a notable, intriguing outlier observed in our ice-velocity records. The reviewer suggests that we remove this section because it "complicates the overall story." By contrast, we would argue that removing outliers—and not attempting to understand these outliers—is a disservice to the observations themselves, and the reality that there are multiple sources and pathways by which meltwater is introduced to the bed producing transient speed-up events captured by ice-sheet velocity records in the ablation zone.

Transient speed-up events captured by ice-velocity records in inland ice-sheet settings have multiple sources: local lake drainages, melt events, precipitation events, *and* propagating subglacial flood events from distal, up-subglacial-catchment lake drainages. The "DOY 180 event" we describe in Section 4.2 falls within this final category, and we would be short-sighted to approach our analysis of transient speed-up events without making the effort to attribute each speed-up event observed in the velocity records to their most likely source. In the first round of revision, we modified the manuscript better motivate this section and to provide a smoother transition from Section 4.1 into this section.

**Specific comments:**

7. Title: Seasonal subglacial drainage system evolution? The authors talk about surface drainage too, but I wonder as the focus is on the inferred evolution of the subglacial drainage system it is worth mentioning this in the title.

We elect to keep the title as is. That we are investigating the subglacial drainage system is indicated with "drainage-system evolution *beneath* the Greenland Ice Sheet"; adding "subglacial" to this title would be redundant.

8. L35: ...(GPS) observations of ice motion show that...
Replaced "Western Greenland Ice Sheet" with "ice motion" on L36

9. L36: Correct van de Wal et al. 2008 reference. I also recommend diversifying references for this bit (e.g., Andrews et al., 2014; Bartholomew et al., 2011)

Revised "del" to "de" on L37 Added references on L38.

10. L38: Delete "the details of". Change "ice-sheet velocity" to "ice velocity" Deleted and revised text on L38.

11. L39: Add "is often non-linear"

Added "often" on L39

12. L44: Suggest changing the end of this sentence to something similar to: "...varies throughout the melt season as subglacial drainage transitions from inefficient to efficient systems, modulating basal sliding. + references"

We elect to modify the suggested revision of this sentence on L49 to: "...and varies throughout the melt season (refs), as subglacial-drainage efficiency evolves and modulates basal sliding (refs)."

13. L46: Change "basal" to "ice"

Revised on L51

14. L48: Change "...how the ice sheet responds to..." to "how ice velocities respond to...". Change "ice-sheet sliding" to "basal sliding".

Revised on L53-54

15. L51: Please add example references for supraglacial lake drainage studies Added on L56-57

16. L52: Supraglacial lakes aren't limited to the western margin, change to/or similar "In the ablation zone of the Greenland Ice Sheet..."

Edited L57

17. L56: Remove "glacial". Change to "reduces friction between the ice and bedrock.."

Revised L61

18. L57: Delete "the" from "the lake drainage events...". Delete "these". Change to "...coincide with surface uplift driven by high water pressures in the subglacial drainage system"

Revised L63 accordingly

19. L59: I might be being pedantic here, but to me "ice sheet" refers to ice sheet wide processes, whereas lake drainages are local/regional. Perhaps say "ice"? Revised to "ice" on L65

20. L62: The added definition is much appreciated here, perhaps change to saturated layer thickness at the ice-bed interface?

Added on L68

21. L68: Change to "surface uplift"

Revised on L74

22. L71: Change to "subglacial drainage efficiency" Added "drainage" to L77

23. L75: Basal channels more commonly refer to channels under ice shelves. Suggest rephrasing sentence to/or similar: "Ice thickness also plays a role, with subglacial

channels under thick ice (define thickness) closing quickly (within hours to days) through ice creep..."

Revised L81-82 accordingly for the 1000-m ice thickness that is relevant to our study area and many other ice-sheet lake-drainage settings.

- 24. L78: Delete "These observations highlight the need for further study on the evolution of basal conditions." Or change "basal conditions"
  Deleted L84
- 25. L81: "ice-sheet speed up" suggests they are occurring over the whole ice sheet. Perhaps delete "ice-sheet". And same for L82?

  Deleted "ice sheet" on L86 and L87
- 26. L100: Suggest adding "localised lake drainage". I'm still not entirely convinced you can confidently state that melt and rainfall events are smaller than lake drainage. With melt/rainfall events happening on much larger spatial scales, the increase in subglacial discharge for well-defined outlets will surely be larger than lake drainages. I suggest instead emphasizing the different spatial scales (local vs regional).

  Added "localized" to L113
- 27. L101: Suggest changing to "transient ice velocity response to meltwater inputs...for annal ice motion". I would suggest refraining/being more careful about the use of "ice-sheet velocities" throughout, with the studies and processes you discuss in this study are all regional/local scales.

Revised L114-115 as suggested

- 28. L103: Again, remove "ice-sheet" Removed "sheet" on L116
- 29. L108-109: Suggest removing this line, as these two types of events operate on vastly different spatial scales.

We elect to keep this sentence. The spatial scale of late-season melt events and precipitation events are inherently of the same spatial scale because both events derive their spatial scales from the scale of atmospheric pressure systems moving over the ice sheet. The horizontal length scales of high- and low-pressure systems are tens-of-kilometers, to a couple hundred kilometers, in diameter. As such, both late-season melt events and precipitation events create runoff across tens-of-kilometer regions of the ice-sheet surface, and both can be considered as "catchment-wide" runoff events.

- 30. L106-120: Good justification for the study, and improved intro to the study site. Thank you!
- 31. L113: Is the site 25 km away from the terminus of Jakobshavn Isbrae, or another glacier? Revised to include that the study site is 25 km up-ice-flow from Saqqarliup sermia, which is a medium-sized tidewater outlet south of Sermeq Kujalleq (Jakobshavn-Isbrae).
- 32. L113: Change "ice-sheet" to "ice motion" Revised on L127
- 33. L139: Suggest rewording 1-sigma errors to 1-standard deviation?

  We elect to keep the wording as is. Reporting formal, 1-sigma errors is the standard error-reporting methodology and wording for position estimates from TRACK. This wording aligns with previous error reporting of these observations in Stevens et al. (2015, 2016, and 2024).
- 34. L223: Are you using the daily mean runoff or daily sum runoff from RACMO?

  Mean added to L219

- 35. L224: Add "ice surface catchment in which..."

  Revised L220
- 36. L226: Shouldn't this be the sum of all grid cells? How many grid cells is the catchment? Revised L223 to state that we averaged across 6 grid cells. We averaged (instead of summed) because we did not think it was realistic that the drainage over the entirety of the basin was responsible for the motion recorded by the North Lake GPS sensors on a given day. Thus, we took an average, to estimate the amount of runoff at any given 11 km x 11 km grid cell (roughly the same scale as the area of the GPS sensors around North Lake) near North Lake as previously described above.
- 37. L228: Change to/or similar "This is a generalized estimate for the runoff that makes it to the bed directly below the lake, but neglects any upstream sources routed beneath North Lake through the subglacial drainage system"

  Revised L223-231 to describe our methodology revision. We use the surrounding RACMO points that could realistically influence the transient speed-up observed at our study area (North Lake) to estimate runoff.
- 38. L235: Change "integrated" to "total" Revised L238
- 39. L254: Again, I don't think these claims can be made without more thorough methods. See previous response (#2)
- 40. L237: Can you please add a reference for the timescale of North Lake drainage? Added Das et al., 2008 and Stevens et al., 2015
- 41. L307: pre-melt season winter velocity? Perhaps for consistency just say "background winter velocity"

Revised to background on L310

- 42. L332: Change "Discussion section" to "Section 4.?" Revised to "Section 4.1" on L337
- 43. L332: The analysis on runoff variability will be more suited here, in the results. See response to #3.
- 44. L336: Suggest changing "ice sheet response" to "ice velocity response" Revised L342
- 45. L329: It will likely not make much difference, but I do think the effective lake drainage melt supply to the system should be your calculated effective runoff + RACMO runoff of the catchment for that day?

Yes, technically that is correct, but we feel it is important to distinguish between RACMO runoff estimates and volume of melt in the lake basin. Given the magnitude of the discrepancy, summing these two quantities does not change the nature of our results. L389: Suggest rephrasing "the sliding behaviour of the Greenland Ice Sheet…" to/or similar "The relationship between ice velocities and surface melt are linked through the evolution of the subglacial drainage system."

Revised on L366

- 46. L396: Change "basal hydrologic system" to "subglacial drainage system" Revised on L379 and L381
- 47. L427: I don't think you can confidently state it is smaller. See previous response #2.
- 48. L434-435: I find this sentence confusing, please can it be rephrased? Rephrased L408-409

- 49. L436: Change "subglacial meltwater system" to "subglacial drainage system" Revised L409
- 50. L448: What is the maximum, mean or integrated flux? Do you mean runoff? Revised to runoff on L428
- 51. L450: Change "speed-up response" to "magnitude of speed-up" or similar Revised L429
- 52. L451: Delete "summer" Deleted on L430
- 53. L452-457: This should be in the methods instead Moved to methods Section 2.2
- 54. L445: This sentence adds confusion, with the "state" of the subglacial drainage system during the melt season being almost completely controlled by the runoff magnitude and variability... (e.g., Schoof, 2010; Hoffman et al., 2011 etc)

As discussed in comment #5, we discuss the subglacial drainage system as distinct from rate of runoff since our results do not show that the velocity response/subglacial efficiency is completely controlled by runoff magnitude and variability The focus of this article is to examine the subglacial system through the dynamic velocity response corresponding to punctuated periods of enhanced melt or flood events. In a broad sense, we agree that runoff is correlated with ice velocities, which we state throughout our introduction and discussion. However, our results do not show a perfect correlation between the magnitude of runoff and velocity response, especially if we consider the large volume entering the system during a lake drainage event.

L450-482: I think this paragraph would be better placed in the results with the other runoff analysis, or possibly replacing the analysis of speed-ups compared to mean, max, etc runoff.

See response to #3. We do not elect to replace the analysis of speed-ups compared to the mean, max, and total event runoff as this highlights the discrepancy between the lake drainage effective runoff vs. RACMO runoff estimates.

- 55. L479: Be careful with the use of "melt events", as I presume you mean peaks in runoff, which could also be caused by rainfall events?
  - Yes, we consider regional melt events and local flood events (associated with lake drainages). As described on L402-405, it is unlikely that the magnitude of precipitation would significantly impact our overall results.
- 56. L484: I suggest rewording to "The correlations between speed-up magnitude, DOY and the rate of change of runoff..."

  Revised on L441
- 57. L485: Change "subglacial drainage state" to subglacial drainage efficiency Revised on L443
- 58. L489: I would suggest rewording/removing. RACMO does not underestimate runoff from lake drainages it doesn't include them at all.

We state on the following L446-447 "RACMO runoff estimates do not account for the meltwater stored in the lake basin." By omitting this line entirely, it does not bring attention to this important consideration to our rate of runoff analysis and results. Thus, the results of Fig S4 underestimate the runoff associated with the DOY 169/162 lake drainage events as plotted.

59. L496-497: Yes, but it is the preceding runoff characteristics that define the "state" of the subglacial drainage system. And if there were a large runoff event before the early-melt season lake drainage, the velocity response would also change.

See response to #5.

60. L508: Delete "glacial"

Deleted on L501

61. L509: Change "ice sheet" to ice

Revised L502

62. L536: Change "ice-sheet" to "ice"

Revised L524

- 63. L542: Confusing sentence, smaller amplitudes in what? Suggest rephrasing "horizontal sliding transients", surely there are no velocity speed-up events before the lake drainage? Revised to "but smaller horizontal sliding transients relative to the pre-speed-up event horizontal sliding" on L530
- 64. L543: remove "the" ...water filled cavities. And network instead of networks? Revised L531
- 65. L545: Change "velocity transient" to "speed-up"

Revised L533

66. L547: Change to: early in the melt season.... Change "hydrologic system" to "subglacial drainage system"

Revised L535

67. L547: Change "the horizontal-velocity increases..." to "Increases in ice velocity associated with..."

Revised L535

- 68. L549: Suggest changing "meltwater conduits" to "connections" Revised L537
- 69. L549: More drainages occur? Or is there simply just more active supraglacial hydrology (moulins etc).

Additional lake drainages occur throughout the melt season resulting in a larger number of cumulative lake drainages as the melt season progresses

- 70. L551: I think this sentence could do with being more nuanced and specifying that it increases frictional coupling in the distributed regions adjacent to efficient channels? Added "adjacent to channels" on L552-553
- 71. L553: Timescale of closure is very dependent on ice thickness
  Sentence revised to: "Finally, late in the melt season, decreased runoff causes channel closing by viscous creep on timescales of hours for the kilometer-scale ice thickness of our study region (Bartholomous et al., 2011), but potentially leaving a network of dewatered cavities."
- 72. L553: Typo? "dewater cavities"

Correct, revised L555 to "dewatered cavities".

- 73. L555: Change conduits to "connections" or just say moulins Revised to moulins on L557
- 74. L631: Change "subglacial bed conditions" to subglacial drainage system Revised L616

75. L632: Suggest removing "preliminary" considering the wealth of previous papers on this site and many other studies using GPS to infer subglacial drainage evolution in west Greenland.

**Removed on L617**

- 76. L645-648: Again, this doesn't fit with the rest of the conclusion/ story on the manuscript See response to #6, which describes why we elect to keep Section 4.2
- 77. L915 (Figure 6): Please check axis labels here and throughout (m year-1 should be m year-1). I think mm/day should also be mm w.e. day-1

  Velocity units revised to "m year-1" in Figures 4, 5, and 10. Runoff units revised to "mm w.e. day-1" throughout.
- 78. Figure 1: Check axis labels throughout (m year-1 should be m year-1). How is the annual ice flow direction is indicated?

  Annual ice-flow direction now included on the map-view panel a with an arrow. The *y*-axis labels for panels that show velocity estimates in this figure already include the correct units for velocity as "m year-1".
- 79. Figure 4: Correct m year-1 Velocity units revised to "m year-1" in Figures 4, 5, and 10.

---

## Author Response (AR3)

**Red Text** – author response stating the edits made to revised version of manuscript (line #s correspond to revised version of manuscript with tracked changes)

**Editor comment:**

I have now also read through the revised manuscript and your response to the reviewer comments. Many thanks for your detailed replies and revisions in response to this. Although I am happy with most of these revisions, the key point raised by the reviewer around the runoff calculation has not been fully addressed i.e. it is typical to sum the runoff values of the individual cells to get the total runoff. The application of a buffer to take account of the travel time of subglacial water makes sense and accounts for your point that not all the melt will be routed to North Lake on the timescales relevant to the observed speed-ups. But, given you now take account of this lag time, it is not clear why you still calculate an average, which then doesn't represent the actual amount of runoff. Please can you either provide stronger justification for calculating an average runoff in your revised methodology, or sum the runoff across the cells within the buffer(s) you use as suggested by the reviewer? Given this is a key part of the method I feel this point needs to be adequately addressed before I can accept the paper.

We acknowledge that our rationale for using an average instead of a summation of runoff feeding the North Lake hydrologic system was not clearly articulated and are happy to provide further clarification.

It is well established that the subglacial hydrologic system moves meltwater beneath an ice sheet following the hydraulic potential (Flowers 2015). During a speed-up event meltwater beneath the lake basin will enter the system both from the surface as well as from the up-stream parts of the subglacial system. Simultaneously meltwater will leave the system to the down-stream parts of the subglacial system. The balance between these fluxes will control the "excess" meltwater in subglacial system, leading to changes in basal effective pressure that modulate the ice sheet response.

This balance is seen clearly during lake drainage events, where a discrete pulse of meltwater from the surficial lake basin enters the subglacial system and then flows outward on timescales of hours to days (Stevens et al., 2015; Lai et al., 2021). As the meltwater pulse migrates away from the lake basin the dynamic response to the event decays. A similar balance occurs during runoff / melt events, where meltwater entering the system is balanced by meltwater moving away from the lake basin, down the hydraulic gradient toward the ice margin (Chandler et al., 2013). A key variable controlling this balance is the rate at which meltwater is supplied to the subglacial system relative to the rate it moves away (Schoof 2010; Hewitt 2013).

Our approach of averaging the runoff rates near the lake is designed to capture the best estimate for the rate of meltwater influx during the speed-up event. The reviewer's argument for integrating this runoff into a volume estimate would make sense only if all the meltwater in this region migrated beneath the lake basin and then remained there for the entire duration of the speed-up event. However, this is unlikely to be the case given that speed-up events last for several days, during which time meltwater will be constantly entering and exiting the hydrologic system below the lake basin. For example, during lake drainage events (when meltwater influx is focused beneath the lake basin on timescales of ~1 hour) there is a clear signal of outward migration of meltwater away from the primary drainage conduit on timescales < 1 day (Stevens et al., 2015). For this reason, we argue that looking at the runoff rate as a proxy for the rate meltwater enters the system is more important than the total volume of meltwater supplied over the speedup event. We take the average of a six grid-cell area because this provides a better representation of the region over which runoff changes will influence the rate meltwater enters the subglacial system beneath the North Lake GPS array.

We stress that in our previous revisions we adopted the reviewer's suggestion to quantify the speed-up signal relative to the rate of change in runoff compared to the pre-event time period. This was an excellent

suggestion as it provides a better proxy for how the rate of inflow to the subglacial system is changing the inflow/outflow balance over the timescale of a speed-up event.

To better highlight out rationale we have modified lines 190–193 and 204–208 of the methods section. If you feel additional clarifications are needed, we would be welcome to suggestions for where to make additional changes to the text.

---

## Author Response (AR5)

**Editor Review:**

The third reviewer has now submitted their report. I tasked the reviewer with evaluating the following statement: it is typical to sum the runoff values of the individual cells to get the total runoff? You will see that overall, they agree with the initial reviewer comment and my concern about how you quantify the runoff. Therefore although I list as minor revisions as it relates just to this one point, I ask that you please better account for how the runoff is calculated following the suggestion(s) of this and the previous reviewer.

Following the third reviewer's suggestion, we revised the runoff methodology to integrate the total melt flux over the supraglacial drainage region feeding the North Lake basin. Specifically, we now calculate the runoff across six local 11-km x 11-km RACMO grid cells to North Lake within the ice-surface catchment basin. The cells included in this area summation were chosen to account for the runoff that makes it to the lake basin on the time scale of a single speed-up event (median event length is 5 days). Following the reviewer's suggestion, we take estimates of open channel supraglacial meltwater flow velocities to determine the spatial extent of the catchment feeding the lake basin. Assuming an open flow velocity of 0.1 m/s or 8,640 m/day (Yang et al., 2018), we calculate that during a single 5-day speed-up event, melt can flow up to distances of ~43 km through the supraglacial system.

To determine which RACMO cells contributed runoff we examined supraglacial streams mapped around North Lake (Joughin et al., 2013). However, these data only extend ~30 km upstream of North Lake (Fig. S3). To identify additional RACMO cells farther upstream that may contribute runoff to North Lake, we first calculated the average sinuosity of the local upstream supraglacial streams in the Joughin et al. (2013) dataset. Taking the ratio of stream length to the straight-line distance from the stream's start to end point we find an average sinuosity of 1.14. Using this value, we calculated the maximum straight-line distance of flow transport to be 37.5 km and included all upstream RACMO cells within this distance from North Lake (Fig. S3). This analysis indicates that six RACMO cells likely feed the North Lake basin on the timescale of single speed-up event. Summing the runoff across these six local grid cells provides an estimate of the total volume of runoff driving transient acceleration at this site (Fig 1d,e). We also estimated the maximum daily run-off rate experienced by the lake basin over the course of a speed up event. To calculate the maximum rate, we used the runoff over a single RACMO grid cell, which most closely corresponds to the distance meltwater can travel in a single day (Fig. S3).

We hope that this approach will address the issue of summing the runoff over the supraglacial drainage basin.

**Referee #3 Review:**

I am tasked with evaluating the following statement by the editor: it is typical to sum the runoff values of the individual cells to get the total runoff?

This is important to the main conclusions of the manuscript, which are (to borrow a reviewer's words) that "there is no relationship between the magnitude of runoff and amplitude of speed-up events" (also a central statement in the abstract). If this statement is to be accurately evaluated, it is important to accurately quantify the runoff driving these speed-ups. As the authors point out, "effective runoff" coming from lake drainages is accurately quantified by measuring the water volume in the lake and its drainage time. Perhaps counterintuitively, it's the day-to-day runoff that is more loosely quantified here, because there is a travel delay between the production time of the runoff and its arrival to the moulin / subglacial system.

The reviewer initially proposed an overestimate of the runoff reaching the moulin / subglacial system during the late-season event (summing all grid cells in the large catchment), which is too high because of delays in surface routing, as the authors point out. But the authors counter with an underestimate (averaging runoff, amounting to allowing input from only one grid cell, sized 11 x 11 km). This is too low because over a 5-

day\* melt event, supposing 0.5 m/s flow velocity in a river, melt from ~50 km away could reach the moulin. Of course, much of the melt has to make its way \_to\_ a river first, which will slow the travel speed considerably. I think that the authors' decision to use the closest ~6 grid cells (travel distance of 20-30 km over 5 days\*) is probably a reasonable compromise. I'd like to see some specific calculation of this distance using known or modeled supraglacial routing travel times or speeds to be sure.

See above response and lines 193–250 of the revised manuscript for a description of how we calculate supraglacial routing. Overall, the resulting catchment region is fully consistent with the reviewer's expectations.

I was confused by the authors' use of the \_subglacial\_ travel speed in their calculation for the distance over which they believe melt could reach the moulin (and thus the subglacial system) during the short runoff event. I can't see how the distance traveled by existing subglacial water would affect this. The authors make a (correct) point that it's the net flux in the subglacial system (output minus input) that's key during an event; however, with the base subglacial outflow assumed to be steady, the only dynamic term here is the surface input. So it's really the \_supraglacial\_ travel speed that should be considered, as I described above.

Following the reviewer's suggestion, we have revised the manuscript to focus on supraglacial flow velocities; however, we note that open channel flow in both supra- and sub- glacial systems have been estimated to be on the same order of magnitude (O 10-1 m/s).

I have reviewed the manuscript in its evolving versions and each review/response. It is still not clear to me why the authors \_average\_ the RACMO input across cells rather than \_summing\_ it across all cells that would be likely to contribute over a ~5-day event. I have to agree with the reviewer that using an average like this is \_not\_ standard practice and that a more careful accounting is necessary. Ideally, this would involve a supraglacial routing routine. This would help the authors avoid another potential pitfall, which is ignoring the runoff in the distant reaches of the catchment produced several days before the event in question, which will be reaching the moulin during the event. I do not think this is large compared to the local runoff generated during the event, but I do think it's likely to be substantial (10-30%).

Here again, we have revised our methodology following the reviewer's recommendation to sum the runoff over the 6 RACMO grid cell area (described above). However, we feel the implementation of a full supraglacial routing algorithm to generate a hydrograph for the North Lake moulin is beyond the scope of this study, as it requires a higher resolution DEM than is available for the 2011 & 2012 melt seasons and a runoff model with higher spatial and temporal resolution.

Since the manuscript's conclusions rely foundationally on the relationship (or lack thereof) between velocity response and runoff, I think the authors need to do a better job quantifying the runoff.

Some other notes I had while reading these materials:

The authors argue that the difference between which cells are included is small, citing Figure S2. However, that figure clearly shows (by my visual estimate) that the 44-cell average method yields runoff about 20%+ larger than the 1- or 6-cell average method. This is not trivial if we are talking about correlations between runoff and velocity response. For comparison, the reviewer's suggestion of the 44-cell sum would be much, much larger than this - 4300%. And a 6-cell total would be 500% larger.

After implementing the reviewer's approach for integrating melt (described above), this issue is no longer relevant in the revised manuscript.

\*I had trouble in the manuscript learning the approximate duration of the runoff events. I found the reviews referring to 5-day events, but I would like to see this timescale reported in the manuscript. Even additional x-ticks in the Figure 1 panels would be helpful.

We have added a supplemental figure (S2) illustrating the duration (days) of each event.